# The Optimal Sine Pulse Frequency of Pulse Hydraulic Fracturing for Reservoir Stimulation

**Heng Li [1], Bingxiang Huang [1],\*  and Hanhua Xu [2],\***

1   State Key Laboratory of Coal Resources and Safe Mining, China University of Mining and Technology, Xuzhou 221116, China
2   Kunming Survey and Design Research Institute of China Nonferrous Metals Industry Co., Ltd., Kunming 650051, China
\*   Correspondence: huangbingxiang@cumt.edu.cn (B.H.); xuhanhua@cug.edu.cn (H.X.)

**Abstract:** Pulse hydraulic fracturing (PHF) is a key technique for reservoir stimulation. PHF can well accelerate the rupture of rock. However, the supercharging mechanism of PHF is not fully understood. The main reason is that the pressure distribution and its variation, especially the peak pressure characteristics, are unclear inside the pipe and fissure. The present research focuses on the sine pulse applied at the inlet of a pipe or fracture to reveal the variation regularity of peak pressure with the pulse frequency, amplitude, pipe length, diameter and wave speed. First, the weakly compressible Navier–Stokes equations were developed to simulate the variation of fluid pressure. The computation codes were developed using the MacCormack method validated by the existing experimental data. Then, the sine pulse effect was studied inside the pipe and fissure. Last, a new frequency model was built to describe the relationship between the optimal pulse frequency, wave speed and pipe length. The results show that there is a family of frequencies at which the peak pressure of the endpoint can be significantly enhanced and that these frequencies are the optimal pulse frequency. It is found that the optimal pulse frequency depends on the pipe or fissure length and wave speed. At the optimal pulse frequency, the peak pressure at the endpoint can be increased by 100% or more, and the cavitation phenomenon occurs. However, the peak pressure decreases when with the decrease in the pipe diameter and fissure departure due to the friction drag effect of the wall. These new landmark findings are very important for the PHF technique. In addition, a new universal frequency model is built to predict the optimal sine pulse frequency. The present research shows the variation regularity of the fluid pressure inside the pipe and develops a sine frequency-controlled method, providing a potential guide for reservoir stimulation.

**Keywords:** fissure flow; pulse hydraulic fracturing; peak pressure; optimal frequency

## 1. Introduction

To enhance the permeability of rock, hydraulic fracturing (HF) is widely used in reservoir stimulation [1–5]. However, the fracturing strength and range are often limited by the finite pressure of hydraulic fluid. As a result, the stimulation effects are not ideal in some situations. Compared to the traditional HF, pulse hydraulic fracturing (PHF) has many advantages and can significantly enhance the fracturing effects [6–9]. However, the related studies are inadequate, especially those touching upon the subjects of peak pressure characteristics of fluid and frequency control in PHF.

There are mainly two reasons why PHF can remarkably accelerate the breakdown of rock. One is that the initiation pressure of rock decreases due to the fatigue load applied by PHF [10–12]. The other is that the peak pressure of fluid increases due to the superposition effect of pressure waves applied by PHF [13,14]. At present, the distribution law of peak pressure is unclear inside the pipe and fissure during the PHF. The inner reason is that the superposition law of pressure waves is vague. The superposition of waves and the

distribution of pressure are affected by the pulse parameters and geometric parameters of pipe and fissure. The pulse parameters include frequency and amplitude. The geometric parameters include pipe length and aperture. Some studies discussed the characteristics of pressure propagation and distribution [15–18]. These studies gave valuable results. However, these studies are inadequate. It is necessary to conduct a systematic study to reveal the law of peak pressure affected by the pulse and geometric parameters.

PHF can decrease the initiation pressure of rock. Originally, PHF was used to weaken the hard roof of coal rock. Later, it was used to enhance the permeability of a coalbed for exploiting the coalbed methane. Zhao [19] conducted PHF studies and found that the initiation pressure decreased and the permeability of coal rock was remarkably enhanced. In the previous studies, Huang et al. [2] studied the fracturing effects of coal-rock mass by combining the blasting technology and PHF method. It was found that the fracturing effects were significantly improved when using the blasting technique and PHF. Cheng et al. [20] investigated the weakening of a hard roof by combining the PHF and constant pumping methods. It was found that pre-pulse hydraulic fracturing can greatly weaken the strength of a hard roof and decrease the breakdown pressure of rock. It was revealed that the average block size of coal caving decreased by 42% when the PHF technique was used. Wu et al. [21] investigated the fracture of cracks in PHF experiments. They found that initiation pressure firstly decreased and then increased with the pulse frequency. The initiation pressure is minimum when the pulse frequency is 4 Hz. Xi et al. [11] concluded that the initiation pressure of rock can be reduced by 10~18% if a 10~4000 pulse cycle is applied in PHF.

PHF can increase the peak pressure of a fluid. Zhai et al. [14] experimentally investigated the distribution characteristics of pulsing water pressure inside a smooth parallel fissure by applying an approximate sine pulse at the inlet. They found that the peak pressure of fluid reaches a maximum at the endpoint of the fissure due to the interaction between the incident and reflection waves. Their study revealed the location of the maximum peak-pressure in PHF. However, the influence of fissure aperture and length on peak pressure and the optimal pulse frequency is vague. Li et al. [22] studied the effects of the pulse frequency on the fracture extension in PHF. They concluded that the initiation pressure is smaller at a lower frequency. However, they also pointed out that the fractures may be easily formed at a higher frequency due to the rapid increase in pressure during the PHF. Diaz et al. [23] discussed the effect of pulse frequency. Their pulse periods are all larger than 30 s. Their study is more similar to intermittent fracturing. In addition, the pulse is trapezoidal in the studies given by Diaz et al. [23].

Based on the previous studies mentioned above, it is found that the low frequency can decrease the initiation pressure of rock, but at the expense of long-time PHF, which is not beneficial for rapid rock-breaking. Hence, the low frequency does not represent the easiness of rock rupture [22]. In other words, the optimal frequency may not be the low frequency. Of course, too-high frequency may not be the most ideal choice for the optimal PHF. Therefore, it is valuable to conduct detailed research to find the optimal frequency in PHF. In addition, most of the past studies were conducted in the true triaxial testing system (TTTS), in which the migration distance of hydraulic fluid is very limited [24,25]. Clearly, the limited flow distance of fluid in TTTS cannot reflect the actual propagation distance of pulse pressure waves in real PHF projects, meaning that the frequency identified in TTTS cannot be applied to actual PHF projects. Therefore, it is extremely necessary to consider the lengths of the pipe and fissure in the PHF. In view of the difficulty of field tests of PHF, numerical simulation is a good choice for revealing the frequency effect on the peak pressure of a hydraulic fluid. In a numerical simulation, the lengths of the pipe and the fissure can be easily considered.

There are two main ways of pressure loading in the PHF. The first way is that the fluid pressure monotonously increases, and then the fluid pressure periodically pulses around a constant pressure until the breakdown of the rock occurs [14]. The second way is that the fluid pressure intermittently increases in a pulsing way until the fracture of the rock

occurs [26]. For example, the peak pressure of each cycle is enhanced with an increment of 10% [26] In the present research, we adopt the first loading way.

Although supercharging phenomena have been reported by some researchers [14], the enhancement of peak pressure is very limited. Hence, it is valuable to further enhance the peak pressure of the fluid to improve the fracturing effects. Although the pulse frequency has been studied [23], the optimal pulse frequency has not been found up to now. Therefore, it is necessary to accurately find the optimal pulse frequency in PHF. Our recent study [27] showed that the square pulse can significantly enhance the fluid pressure near the pipe endpoint. However, the influence of the sine pulse is unclear. Objectively, the variations of the fluid pressure are different in using the square and sine pulse injections. Therefore, it is necessary to discuss the topic of sine pulse injection in this paper.

In this article, we focus on the issue of the peak pressure of fluid in PHF by adjusting related sine pulse parameters. The main objective is to find the optimal pulse frequency to achieve significant supercharging and reveal the supercharging mechanism. The methods adopted here include theoretical analysis and numerical simulations involving solving the Navier-Stokes (N-S) equations.

## 2. Equation and Method

PHF and water hammer have similar physical characteristics, such as the change in fluid density. In the processes of PHF and water hammer, the fluid density has a slight change due to the pressure difference, so the water hammer equations are usually used to simulate the flow process of PHF [28,29]. Essentially, the water hammer equations are simplified from the N-S equations which include the continuity (mass conservation) equation and momentum equation. Water hammer equations adopt the assumption that the fluid is weakly compressible. The weakly compressible assumption means that the density change is considered in the continuity equation, and the density change is not considered in the momentum equation approximately. The weakly compressible assumption is helpful in the numerical computation because the pressure and velocity are coupled. In addition, the weakly compressible assumption considers the density change to a certain extent and also avoids the complexity of a differential equation. Therefore, the one-dimensional hammer equations are adopted here.

### 2.1. Model Equation

Considering the one-dimensional characteristics of pipe flow in PHF, the weakly compressible N-S equations are given as follows:

$$\begin{cases} \frac{\partial p}{\partial t} + u\frac{\partial p}{\partial x} + \rho a^2 \frac{\partial u}{\partial x} = 0 \\ \frac{\partial u}{\partial t} + \frac{1}{\rho}\frac{\partial p}{\partial x} + u\frac{\partial u}{\partial x} = -\lambda\frac{u|u|}{2d} \end{cases} \tag{1}$$

where $p$ is the fluid pressure, $u$ is the fluid velocity, $\rho$ is the fluid density, $a$ is the wave speed, $d$ is the diameter of the pipe and $\lambda$ is the friction drag coefficient. For the laminar flow, the friction drag coefficient is defined by $\lambda = 64/Re$, where $Re$ is the flow Reynolds number. For the turbulent flow, the friction drag coefficient is defined by $1/\sqrt{\lambda} = 2log\left(Re\sqrt{\lambda}\right) - 0.8$, which is an implicit expression about $\lambda$. This implicit formula cannot be directly solved. The Newton iteration method is used to obtain the approximate solution of the friction drag coefficient. The pressure is related to the water head, and there is the relation $p = \rho gh$. Then, the N-S equations can also be written in the form of water head [28] as follows:

$$\begin{cases} \frac{\partial h}{\partial t} + u\frac{\partial h}{\partial x} + \frac{a^2}{g}\frac{\partial u}{\partial x} = 0 \\ \frac{\partial u}{\partial t} + g\frac{\partial h}{\partial x} + u\frac{\partial u}{\partial x} = -\lambda\frac{u|u|}{2d} \end{cases} \tag{2}$$

where $h$ is the water head. Essentially, Equations (1) and (2) are equivalent. For the sake of numerical calculation, Equation (2) is usually rewritten as

$$\begin{cases} \frac{\partial h}{\partial t} = -u\frac{\partial h}{\partial x} - \frac{a^2}{g}\frac{\partial u}{\partial x} \\ \frac{\partial u}{\partial t} = -g\frac{\partial h}{\partial x} - u\frac{\partial u}{\partial x} - \lambda\frac{u|u|}{2d} \end{cases} \tag{3}$$

### 2.2. Numerical Computation Method

For the simulation of weakly compressible flow, the common methods include the property line method and one-order finite difference. These methods have some disadvantages such as limits of smaller time-step or lower accuracy. The MacCormack method [30] has two-order accuracy in time and space solutions and is a good choice for the simulation of PHF. The main idea of the MacCormack method is firstly predicting the pressure and velocity at time-step $t+\triangle t$, then solving the corrected pressure and velocity at time-step $t+\triangle t$, and then repeating this procedure. The differential equation can be written in the discrete form given by Equation (4):

$$\begin{cases} \left(\frac{\partial h}{\partial t}\right)_i^t = -u_i^t\frac{h_{i+1}^t-h_i^t}{\Delta x_{i+1}} - \frac{a^2}{g}\frac{u_{i+1}^t-u_i^t}{\Delta x_{i+1}} \\ \left(\frac{\partial u}{\partial t}\right)_i^t = -g\frac{h_{i+1}^t-h_i^t}{\Delta x_{i+1}} - \overline{u}_i^t\frac{u_{i+1}^t-u_i^t}{\Delta x_{i+1}} - \lambda\frac{u_i^t|u_i^t|}{2d} \end{cases} \tag{4}$$

The time derivative term is discretized into the difference form including the predicting variables as follows:

$$\begin{cases} \frac{\overline{h}_i^{t+\Delta t}-h_i^t}{\Delta t} = -u_i^t\frac{h_{i+1}^t-h_i^t}{\Delta x_{i+1}} - \frac{a^2}{g}\frac{u_{i+1}^t-u_i^t}{\Delta x_{i+1}} \\ \frac{\overline{u}_i^{t+\Delta t}-u_i^t}{\Delta t} = -g\frac{h_{i+1}^t-h_i^t}{\Delta x_{i+1}} - \overline{u}_i^t\frac{u_{i+1}^t-u_i^t}{\Delta x_{i+1}} - \lambda\frac{u_i^t|u_i^t|}{2D} \end{cases} \tag{5}$$

Based on the above discrete equations, the prediction variables $\overline{h}_i^{t+\Delta t}$ and $\overline{u}_i^{t+\Delta t}$ are obtained. The predicting derivative terms are given by Equation (6):

$$\begin{cases} \left(\frac{\partial h}{\partial t}\right)_i^{t+\Delta t} = -\overline{u}_i^t\frac{\overline{h}_i^{t+\Delta t}-\overline{h}_{i-1}^{t+\Delta t}}{\Delta x_i} - \frac{a^2}{g}\frac{\overline{u}_i^{t+\Delta t}-\overline{u}_{i-1}^{t+\Delta t}}{\Delta x_i} \\ \left(\frac{\partial u}{\partial t}\right)_i^{t+\Delta t} = -g\frac{\overline{h}_i^{t+\Delta t}-\overline{h}_{i-1}^{t+\Delta t}}{\Delta x_i} - \overline{u}_i^t\frac{\overline{u}_i^{t+\Delta t}-\overline{u}_{i-1}^{t+\Delta t}}{\Delta x_i} - \lambda\frac{\overline{u}_i^{t+\Delta t}|\overline{u}_i^{t+\Delta t}|}{2D} \end{cases} \tag{6}$$

Finally, the corrected water head and velocity are obtained as follows:

$$\begin{aligned} h_i^{t+\Delta t} &= h_i^t + \frac{\Delta t}{2}\left(\left(\frac{\partial h}{\partial t}\right)_i^t + \left(\frac{\partial h}{\partial t}\right)_i^{t+\Delta t}\right) \\ u_i^{t+\Delta t} &= u_i^t + \frac{\Delta t}{2}\left(\left(\frac{\partial u}{\partial t}\right)_i^t + \left(\frac{\partial u}{\partial t}\right)_i^{t+\Delta t}\right) \end{aligned} \tag{7}$$

### 2.3. Boundary Condition

It is vital to define the boundary conditions. At the inlet, the commonly used boundary conditions include pressure boundary and velocity boundary. At the outlet, the commonly used boundary includes pressure boundary, velocity boundary and wall boundary. At the wall, the no-slip wall condition is widely used. For the simulation of water hammer, the pressure boundary condition is adopted at the inlet, and the wall boundary is used at the outlet when the valve closes. The no-slip wall condition is adopted at the pipe wall. The left inlet boundary conditions [28] are given as

$$\begin{cases} h_0 = Zu \\ u_0 = -a^2u_1/\left[g(h_1-h_0) - a^2\right] \end{cases} \tag{8}$$

where $Zu$ is the constant water head at the inlet. The velocity boundary condition is derived by $h_0 = Zu$, namely $(\partial h / \partial t)_0 = 0$. The right outlet boundary condition [28] is given as

$$\begin{cases} h_{nx+1} = h_{nx+1}^t + \Delta t a^2 u_{nx} / (g \Delta x_{nx}) \\ u_{nx+1} = 0 \end{cases} \tag{9}$$

It is noteworthy that the pressure boundary condition is derived by $u_{nx+1} = 0$, namely $(\partial u / \partial t)_{nx+1} = 0$.

For the simulation of PHF, the pressure boundary condition is adopted at the inlet and continuous sine pulse pressure is applied at the inlet. Because the endpoint is blind, the wall boundary is used at the endpoint where the velocity and pressure satisfy the reflection condition of the wall. The no-slip wall condition is adopted at the pipe wall. The left inlet condition is given as follows:

$$\begin{cases} h_0 = A_p sin(2\pi f \cdot t) + h_{av} \\ u_0 = -\dfrac{a^2 u_1}{g(h_1 - h_0) - a^2} - \dfrac{g \Delta x \cdot 2\pi f A_p cos(2\pi f \cdot t)}{g(h_1 - h_0) - a^2} \end{cases} \tag{10}$$

where the velocity boundary condition is derived by combining $\left( \frac{\partial h}{\partial t} \right)_0 = 2\pi f \cdot A_p cos(2\pi f \cdot t)$ and the continuity mass equation. It is noteworthy that the water head $h$ represents the normalized fluid pressure according to the relation $h = p/(\rho g)$. Hence, the pressure and velocity boundary conditions have been given in Equation (10). The right outlet condition is

$$\begin{cases} h_{nx+1} = h_{nx+1}^t + \Delta t a^2 u_{nx} / (g \Delta x_{nx}) \\ u_{nx+1} = 0 \end{cases} \tag{11}$$

The outlet boundary condition is

$$\begin{cases} h_{nx+1} = h_{nx} \\ u_{nx+1} = -u_{nx} \end{cases} \tag{12}$$

Based on the above equations, algorithm and boundary conditions, we developed the calculation program using Fortran language. The validation of the present program is shown in the following section in detail. In PHF, a simplified physical model is adopted as shown in Figure 1. The pipe length is $L_x$, the pipe inner diameter is $d$, the left end is the inlet and the right endpoint is the blind end. In realistic PHF, first, the steel pipe is embedded into the rock formation before the injection of a hydraulic fluid such as water. Then, the water is injected into the steel pipe and the water pressure gradually increases to a particular value such as $p = 1$ MPa, which is less than the initiation pressure of rock. Usually, the initiation pressure of rock is between 5 MPa and 20 MPa [24,31,32]. Last, the pulse pressure is periodically applied at the inlet by the pulse pump. In this article, we focus on the loading stage of pulse pressure before the breakdown of rock. In fact, the loading stage of pressure is an important stage before the rupture of rock. Therefore, the present work is meaningful. The supercharging process is studied in detail at the endpoint of the pipe. The present studies and findings are valuable for the PHF design.

## 2.4. Validation of Experiments

Objectively, there is strong similarity between the PHF and water hammer, where all of the pressure excitations propagate in the form of waves in the weakly compressible medium of water. Therefore, the present algorithm and program are firstly validated by two classical water hammer experiments given by Bergant et al. (2001) [28]. In Brunone's model [33], the friction drag coefficient considers the influence of transient acceleration and convective acceleration. To maintain consistency with Brunone's experiments [33], we also consider the influence of transient and convective acceleration by correcting the friction drag coefficient.

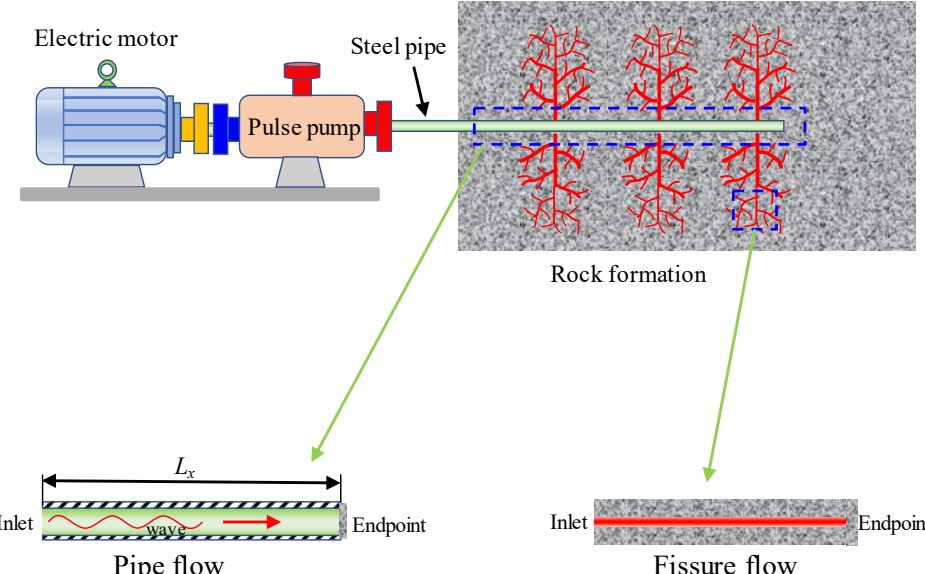

**Figure 1.** Diagram of the flow model in pipe and fissures during PHF; the upper part is the schematics of PHF and fissure networks, and the lower part is the local enlargement of pipe and fissure, whose length is $L_x$ and width is $d$. Pulse pressure wave is applied by the pulse pump.

The first experiment focused on the laminar flow case at Reynolds number $Re = 1870$ inside a pipe, whose length is 37.23 m and internal diameter is 2.21 cm. The initial static head at the inlet is 32 m, which remains constant during the experimental process. The initial flow velocity is $V_0 = 0.1$ m/s, and the valve closure time is at $t = 0.009$ s. The wave speed of the water hammer is $a = 1319$ m/s when the water temperature is 15.4 °C.

Results are shown in Figure 2. It can be seen that the water head (or pressure) periodically fluctuates at the middle point and endpoint due to the reflection of pressure waves. However, the peak of the water head gradually decreases due to the friction drag effect of the wall. The present computation results agree well with the experimental results [28]. Particularly, the present peak and frequency of the water head given by the numerical computation have good consistency with the experimental data, which indicates that the present algorithm and calculation program can give accurate simulation results.

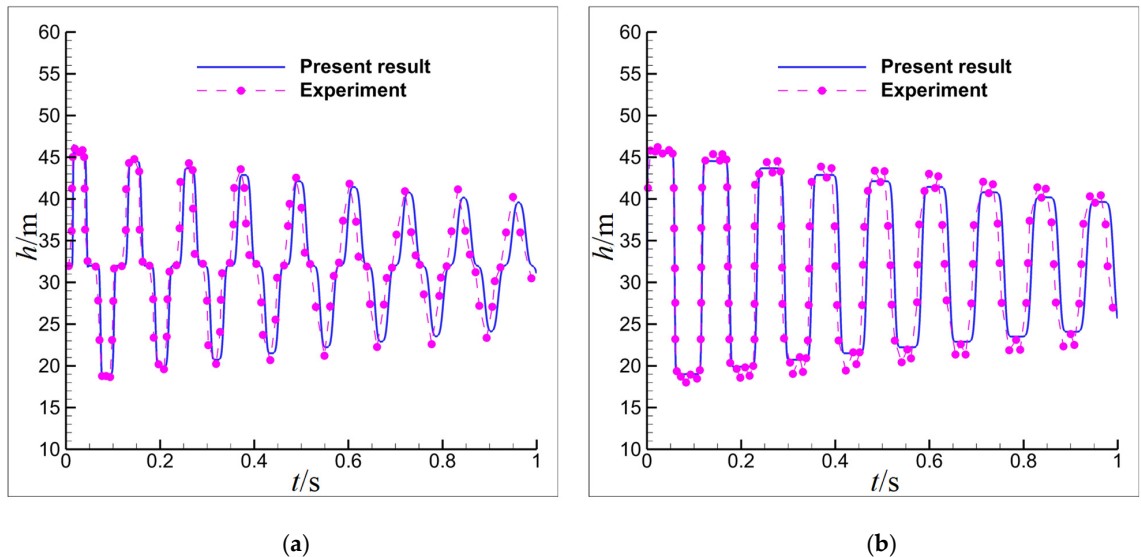

**Figure 2.** Comparison of water heads between the present simulation and experimental data when $V_0 = 0.1$ m/s: (**a**) at the midpoint, (**b**) at the valve (endpoint).

The second experiment was also conducted in the above pipe with the same size, including the same length and diameter. However, the initial flow velocity is larger than that in the first case. The flow velocity is 0.3 m/s, and the Reynolds number is 5600. In this case, the flow is turbulent instead of laminar. The simulation is conducted based on these parameters. The results are shown in Figure 3. It is found that the present numerical results agree well with the experimental data [28]. These comparisons further confirm that the present calculation program is accurate and credible.

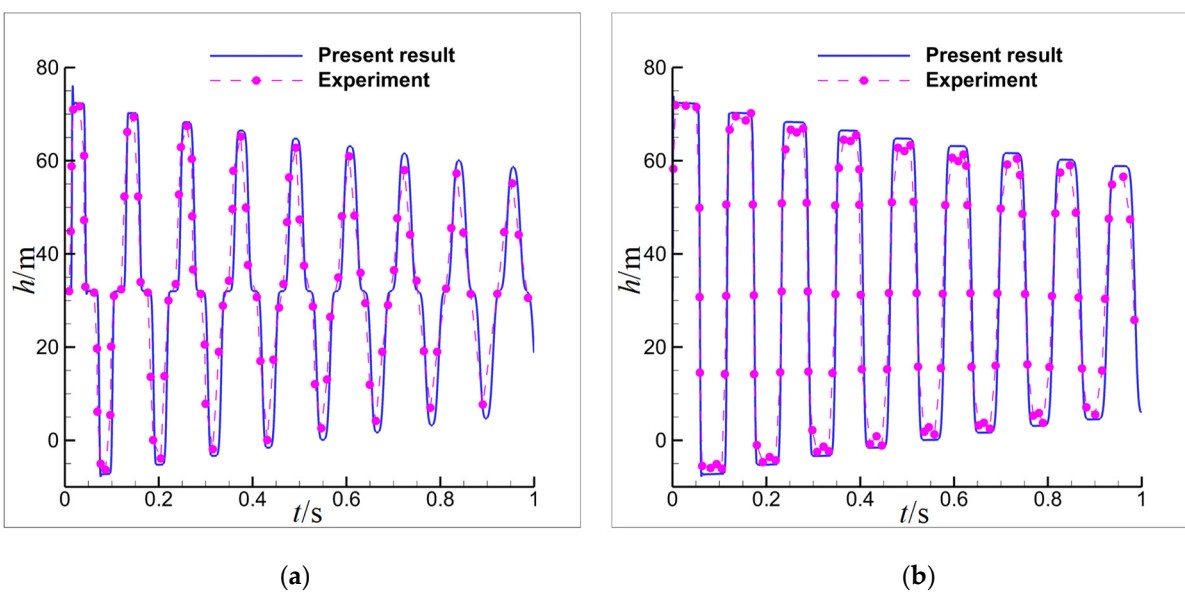

(**a**)  (**b**)

**Figure 3.** Comparison of water heads between the present simulation and experimental data when $V_0 = 0.3$ m/s: (**a**) at the midpoint, (**b**) at the valve (endpoint).

## 3. Results and Discussion

### 3.1. Pressure Characteristic of Fluid when Ignoring Drag Effect of Wall

For the low-velocity flow inside a pipe with a larger diameter, the friction drag is usually negligible. In this part, we consider an ideal situation where the drag effect of the wall is ignored during the propagation of the pulse pressure wave. It is vital to conduct this study in an ideal situation, which helps to reveal the basic propagation law of pulse pressure and lay the foundation for the research on PHF in real pipes and fissures. In the ideal situation, the drag effect is ignored and the control equation can be written as

$$\begin{cases} \frac{\partial p}{\partial t} + u\frac{\partial p}{\partial x} + \rho a^2 \frac{\partial u}{\partial x} = 0 \\ \frac{\partial u}{\partial t} + \frac{1}{\rho}\frac{\partial p}{\partial x} + u\frac{\partial u}{\partial x} = 0 \end{cases} \tag{13}$$

where $p$ is the fluid pressure. The normalized pressure $P_n$ is represented by $P_n = p/p_{av}$, where $p_{av}$ is the average pressure of the sine pulse at the inlet. Essentially, $P_n$ represents the magnification of pressure, which can be used to describe the magnification level of fluid pressure from the inlet ($p_{av}$) to any position ($p$). In the following analysis, the maximum peak pressure is expressed by $P_{n,max}$, and the minimum peak pressure is expressed by $P_{n,min}$, as shown in Figure 4.

To solve the control equation, the MacCormack method is used here, and this method has been introduced above. In addition, it is vital to define the initial and boundary conditions to obtain the numerical solution. The initial conditions include that the fluid pressure equals the average pressure of the sine pulse and that the fluid velocity is zero. The boundary conditions include that the consecutive sine-pulse pressure is periodically applied at the inlet of the pipe and that the endpoint is enclosed. In other words, the endpoint is a blind end at which the velocity and pressure satisfy the reflecting boundary

condition. To ensure that the numerical results do not depend on the grid, grid-independent tests are conducted at first. It is confirmed that 500 uniform grid points are enough for the following simulations, so 500 computation points are adopted in the later PHF simulations.

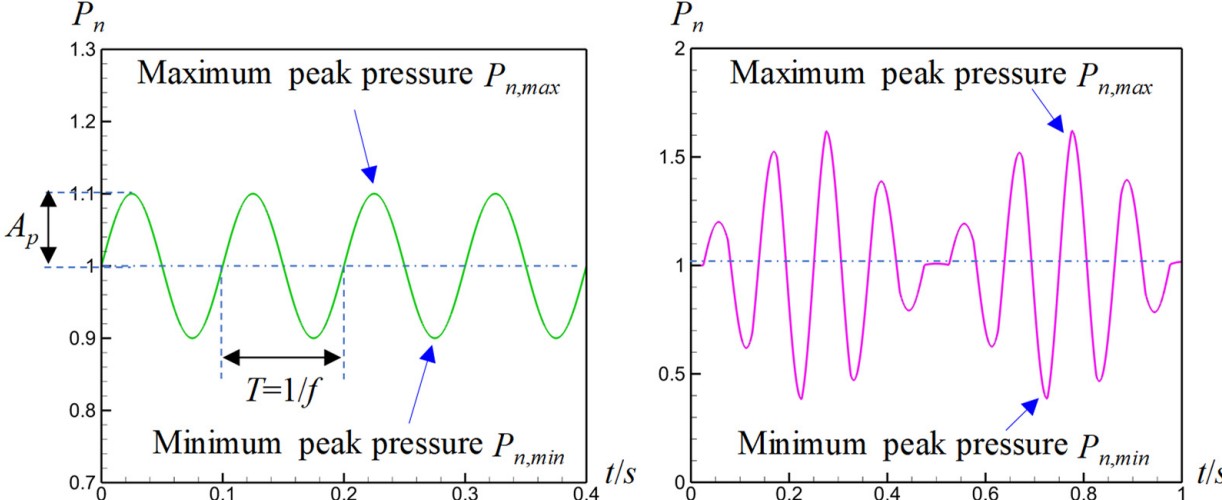

**Figure 4.** Sketch map of normalized pressure $P_n$, where $P_{n,max}$ is the maximum peak pressure and $P_{n,min}$ is the minimum peak pressure. The left part is the sine pulse applied at the inlet. The right part is the pressure–time curve somewhere in the pipe. All normalized pressures are dimensionless quantities.

The fluid pressure is affected by the reflecting wave, which is associated with the endpoint location. Therefore, the pipe length should be given first. Here, four lengths are adopted, namely $L_x$ = 5 m, 10 m, 25 m and 250 m, covering a wide range of pipe lengths. The wave speed is set as 1000 m/s.

The pulse frequency is another important parameter affecting the propagation of waves and distribution of pressure inside the pipe or crack. A series of frequencies are considered in the present study from low to high. The frequency ranges from $f$ = 1 Hz to $f$ = 100 Hz. The peak pressure usually appears at the endpoint of the pipe or crack when the wall drag effect is inappreciable. The pressure–time curves are shown in Figure 5.

At low frequency $f$ = 1 Hz, the fluid pressure at the endpoint is consistent with the sine-pulse pressure at the inlet as shown in Figure 5a,d,g. Their pressure–time curves are almost coincident, which indicates that their amplitudes are similar and their wave periods are the same at the inlet and endpoint. This phenomenon can be explained by the basic law of wave propagation. The wave speed is 1000 m/s, and the inlet pulse disturbance travels to the endpoint only needing 0.01 s for $L_x$ = 10 m and 0.005 s for $L_x$ = 5 m. Obviously, the propagation time is far less than the sine pulse period $T = 1/f$ = 1 s. Therefore, the pressure–time curves are similar at the inlet and endpoint. This phenomenon indicates that low-frequency excitation of the sine pulse could not enhance the peak pressure near the endpoint. In fact, we have examined the whole pressure distribution inside the pipe and found that the amplification effect of pressure is inappreciable. Hence, the conclusion is that low frequency is not a good choice for enhancing the peak pressure of the fluid.

At high frequency $f$ = 100 Hz, the amplification effect of pressure is weak at the endpoint. As shown in Figure 5f,i, the nondimensional peak pressure is $P_n$ = 1.1 at the inlet and $P_n$ = 1.2 at the endpoint, where the amplification is not apparent. This phenomenon indicates that part of the high-frequency pulse provides a very limited amplification effect of pressure. This phenomenon was reexamined in long and short pipes. It is confirmed that the amplification effect of pressure is weak in most cases when the pulse is applied with a higher frequency ($f$ = 100 Hz) at the inlet.

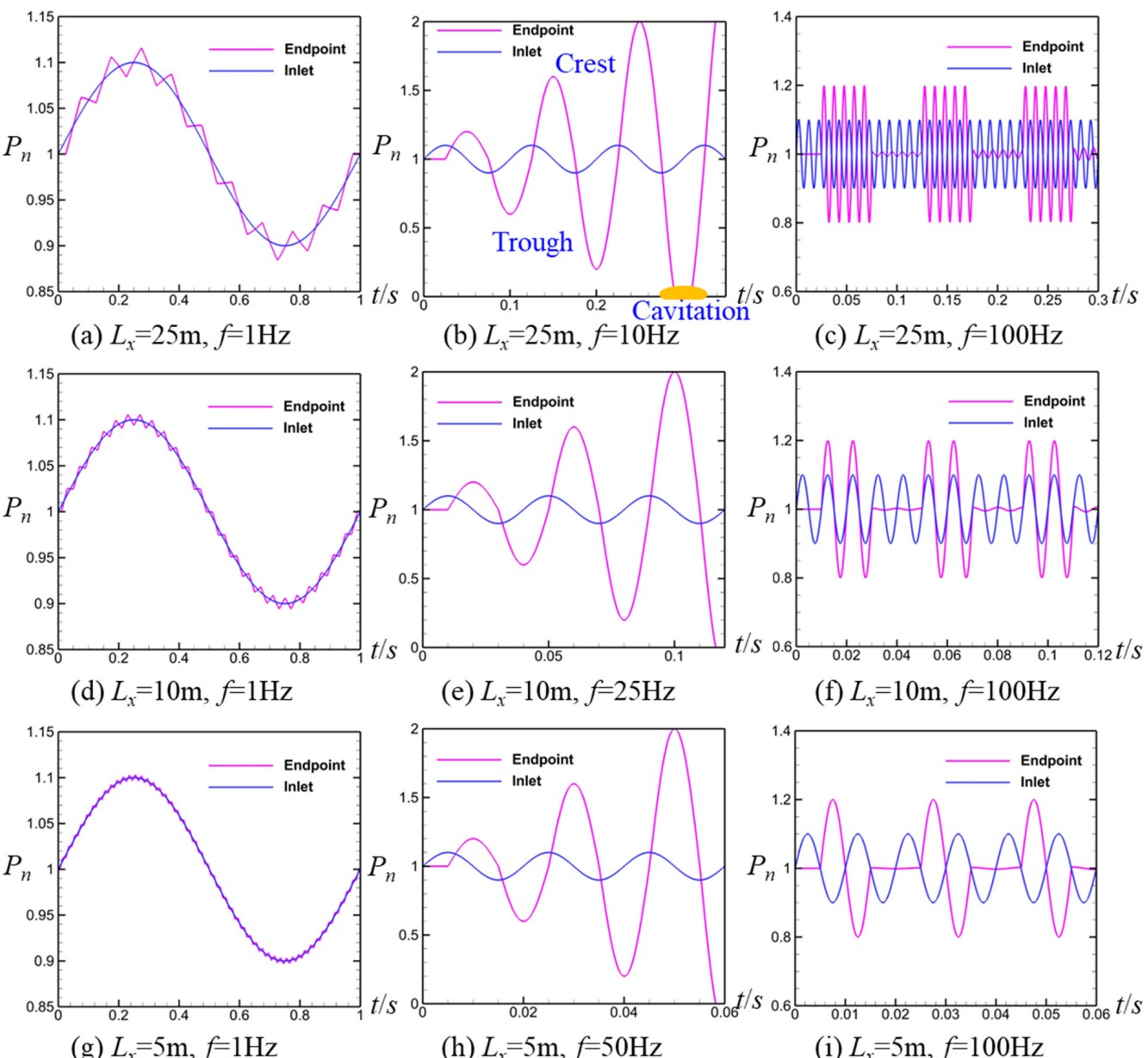

**Figure 5.** Normalized pressure–time $P_n$-$t$ curves at inlet and endpoint: (**a**) $L_x = 25$ m, $f = 1$ Hz; (**b**) $L_x = 25$ m, $f = 10$ Hz; (**c**) $L_x = 25$ m, $f = 100$ Hz; (**d**) $L_x = 10$ m, $f = 1$ Hz; (**e**) $L_x = 10$ m, $f = 25$ Hz; (**f**) $L_x = 10$ m, $f = 100$ Hz; (**g**) $L_x = 5$ m, $f = 1$ Hz; (**h**) $L_x = 5$ m, $f = 50$ Hz; (**i**) $L_x = 5$ m, $f = 100$ Hz. All normalized pressures are dimensionless quantities.

For the middle-frequency excitation of the sine pulse, the amplification effect of pressure is appreciable. As shown in Figure 5b,e,h, the nondimensional peak pressure is magnified up to $P_n = 2$ at the third wave crest (increasing by 100%). This phenomenon indicates that the middle-frequency pulse excitation can well enhance the fluid pressure near the endpoint. We examined the whole pressure distribution inside the pipe and found that the peak pressure of all regions except the inlet is enhanced, especially near the endpoint. This phenomenon can be explained by the consecutive superposition of incident and reflected waves. The premise of effective superposition is that the incident wave is at an 'appropriate' frequency. This appropriate frequency is closely related to the pipe length if the wave speed is constant. In other words, the appropriate frequency should be determined according to the distance between the inlet and the endpoint. Our results show that the optimal frequency is 10 Hz for $L_x = 25$ m. The optimal frequency increases to 25 Hz

when the pipe length decreases to 10 m. The optimal frequency increases to 50 Hz when the pipe length decreases to 5 m. If the pipe length further decreases to 1 m, the optimal frequency increases to 250 Hz. At the optimal pulse frequency, the fluid pressure at the endpoint can be significantly enhanced compared to the initial sine pressure at the inlet.

The root cause of peak-pressure amplification is that the supercharging effect exists due to the reflection and superposition of pressure waves. The sine-pulse waves are continually injected at inlet, and these continuous pressure waves reflect at the endpoint. There is a complex interaction between the incident waves and the reflected waves. As a result, the superposed pressure at the endpoint has a variation period that is different from the initial period of the sine pulse. A more detailed explanation and discussion are given in Section 3.5.

Figure 5b shows that the peak pressure at the endpoint increases with time at the frequency $f$ = 10 Hz. However, this does not mean that the peak pressure can increase to infinity. In fact, the pressure drops to 0 near the third wave trough, indicating that hydrodynamic cavitation must occur. When the fluid pressure drops below the saturation vapor pressure, hydrodynamic cavitation occurs. For example, the saturation vapor pressure is 0.0024 MPa when the water temperature is 20 °C [34]. Our results show that there is hydrodynamic cavitation at the third wave trough (Figure 5b). When the cavitation occurs, the cavitation bubbles appear and develop, and these cavitation bubbles will collapse if the fluid pressure rises again, as occurs during the fourth ascending stage of the pressure wave at the endpoint (Figure 5b). As is known to all, the collapse of cavitation bubbles will cause extremely high pressure, usually larger than 50 MPa, while the fracture pressure of rock is usually between 5 MPa and 20 MPa [24,31,32]. Therefore, the rock rapidly fractures.

In summary, the low-frequency excitation cannot enhance the maximum peak pressure. Part of the high-frequency excitation can enhance the maximum peak pressure but in a very limited way. The middle-frequency excitation can significantly enhance the maximum peak pressure of the fluid. In the following part, we focus on the middle-frequency pulse and discuss its characteristics in detail.

We took $L_x$ = 25 m as an example and chose nine pulse frequencies for testing. As shown in Figure 6, it is found that the endpoint peak pressure is significantly enhanced when the inlet pulse frequency is 9 Hz, 10 Hz and 11 Hz, especially at 10 Hz. At $f$ = 10 Hz, the maximum peak pressure is enhanced by 100% up to $P_n$ = 2 and the cavitation phenomenon is observed at $t$ = 0.29 s. For other frequencies, such as $f$ = 6~8 Hz and $f$ = 12~14 Hz, the magnification of peak pressure is not apparent and no cavitation phenomenon is observed. For example, at $f$ = 6 Hz, the maximum peak pressure at the endpoint is only $P_n$ = 1.3 (shown in Figure 6a,j), so the magnification is not apparent. At $f$ = 6 Hz, the minimum peak pressure at the endpoint is $P_n$ = 0.7, which is higher than the saturation vapor pressure if the reference pressure at the inlet is 1 MPa, so no cavitation phenomenon occurs.

Compared to $f$ = 9 Hz and $f$ = 11 Hz, the frequency $f$ = 10 Hz provides the largest peak pressure before the appearance of the cavitation phenomenon (shown in Figure 6j). On the whole, the frequency of $f$ = 10 Hz gives the best magnification effect compared to other frequencies. Therefore, $f$ = 10 Hz can be identified as the optimal pulse frequency.

Because the sine wave has periodicity, there may be multiple frequencies that belong to the optimal frequency. We tested a series of frequencies from low to high with a small frequency interval. Results show that, indeed, there is a family of frequencies at which the maximum peak pressure is enhanced by 100% up to $P_n$ = 2 as shown in Figure 7. These optimal frequencies include 10 Hz, 30 Hz and 50 Hz. It is found that the minimum peak pressure approaches 0 near these optimal frequencies. Namely, the hydrodynamic cavitation occurs near these frequencies, which can remarkably enhance the local pressure at the endpoint due to the collapse of cavitation bubbles.

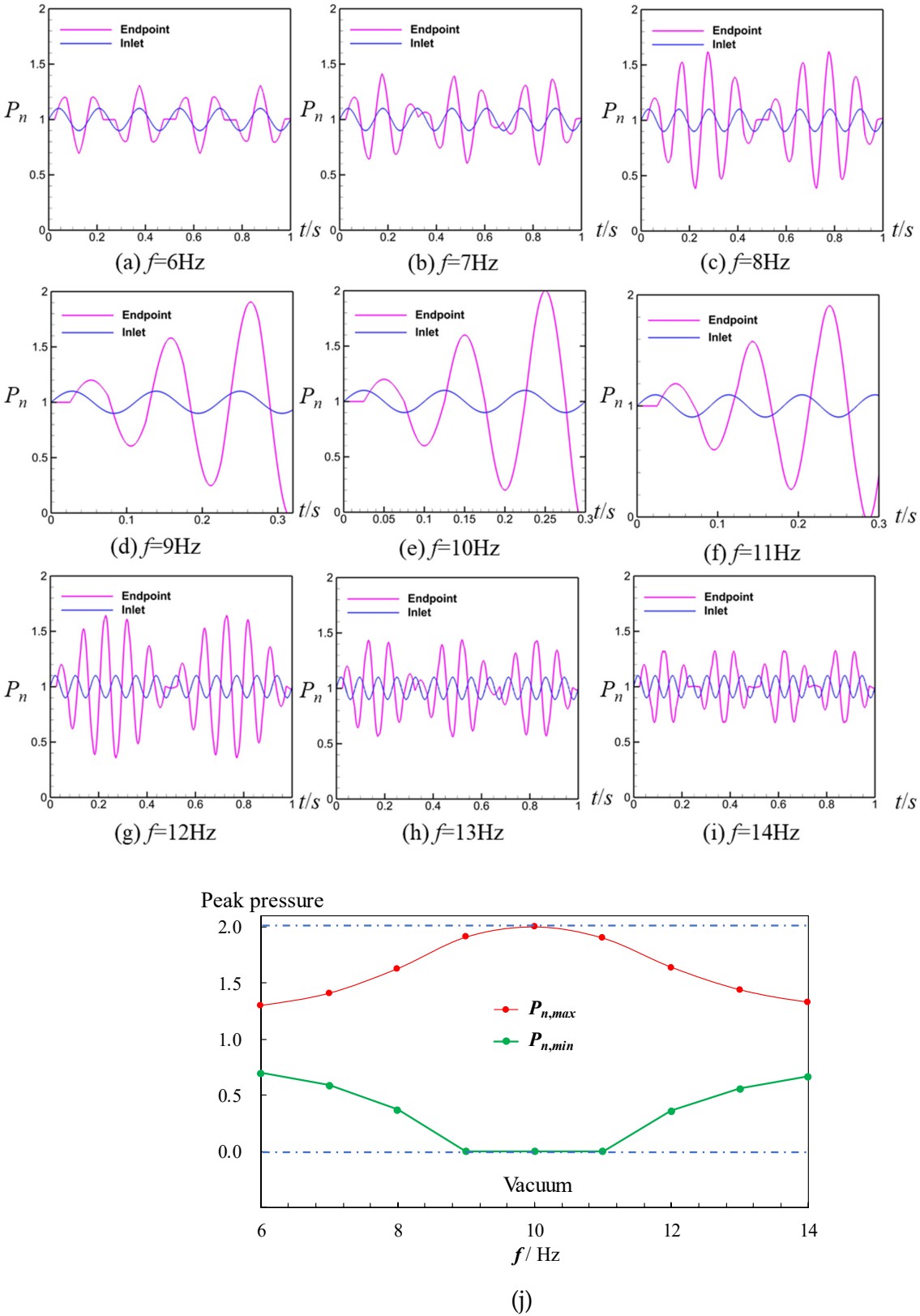

**Figure 6.** Normalized pressure–time $P_n$ curves at inlet and endpoint when $L_x$ = 25 m: (**a**) $f$ = 6 Hz, (**b**) $f$ = 7 Hz, (**c**) $f$ = 8 Hz, (**d**) $f$ = 9 Hz, (**e**) $f$ = 10 Hz, (**f**) $f$ = 11 Hz, (**g**) $f$ = 12 Hz, (**h**) $f$ = 13 Hz, (**i**) $f$ = 14 Hz. (**j**) Relationship between the normalized peak pressure $P_{n,max}$, $P_{n,min}$ and frequency $f$ at endpoint. All normalized pressures are dimensionless quantities.

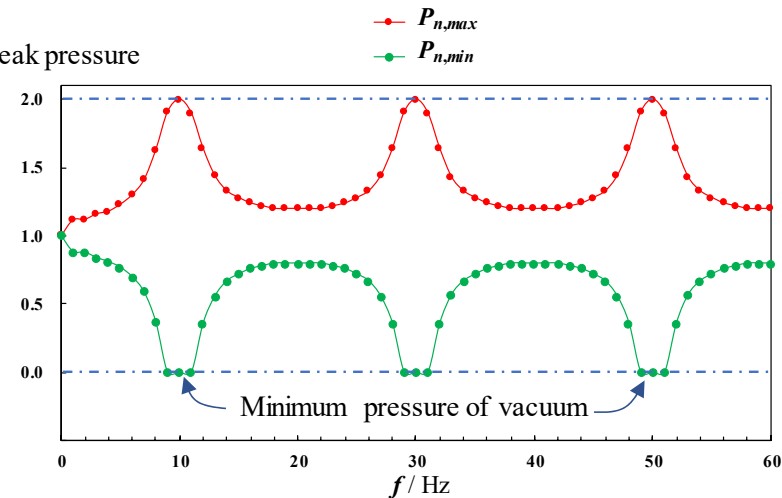

**Figure 7.** Relationship between the normalized peak pressure $P_{n,max}$, $P_{n,min}$ and frequency $f$ at endpoint where $L_x$= 25 m.

The pulse frequency producing the largest $P_{n,max}$ (Figure 7) is defined as the optimal (pulse) frequency, at which the magnification effect is most apparent. The definition of the optimal frequency is given as

$$\begin{cases} P_{n,op} = \boldsymbol{max}\{P_{n,max}(f)\}, f > 0 \\ P_n(f = f_{op}) = P_{n,op} \\ P_n(f \neq f_{op}) < P_{n,op} \end{cases} \tag{14}$$

where $\boldsymbol{max}\{\}$ is a mathematical expression to obtain the maximum from a set of elements, and $f_{op}$ is the optimal pulse frequency at which the fluid pressure can reach the maximum before the hydrodynamic cavitation. For example, one of the optimal pulse frequencies is 10 Hz for the case of $L_x$ = 25 m, as shown in Figures 6 and 7.

To identify the optimal frequency for any pipe, a great number of cases were computed and analyzed from low frequency to high frequency with a small frequency interval. For every case, the corresponding peak pressures are recorded at the endpoint. It is found that most of the maximum peak pressures are enhanced by less than 50% compared to the inlet average pressure. These results are consistent with the previous experimental results [14]. For example, the previous experimental studies showed that the sine pulse could enhance the peak pressure by about 0~20% at the endpoint [14]. In fact, the present research also confirms that most of the pulse frequencies can only provide a less obvious amplified effect. Namely, the peak pressure is enhanced usually by less than 50% for the most of pulse frequencies. That is the reason why the traditional studies only gave 0~20% enhancement by the PHF method and could not give significant amplification such as enhancement by 100% or more. The key is that the traditional studies did not find the optimal pulse frequency $f_{op}$.

To find the optimal pulse frequency, we tested a series of frequencies from low to high under the condition of constant pipe length. Then, by testing a series of pipes from short to long, the corresponding optimal frequencies are obtained. By comparison of long and short pipes, it is found that the optimal frequency decreases with the increase in pipe length as shown in Figure 8. We find that there is a quantitative relation between the optimal frequency and pipe length. Based on the regressive analysis, this quantitative relation is established, and its expression is $f_{op} = k f_0 / L_x$, where =1, 3, 5, . . . , $f_0 = 250$ Hz•m and $L_x$ is the length of pipe. It is noteworthy that this model is built under the assumption of wave speed $a = 1000$ m/s. This model shows that the optimal pulse frequency is inversely proportional to the pipe length (shown in Figure 8).

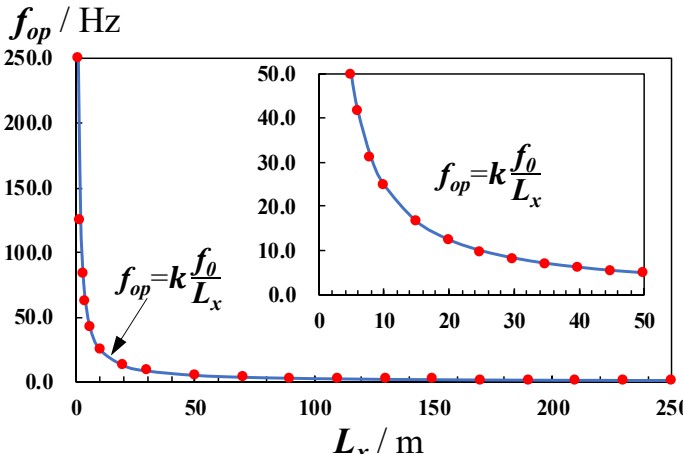
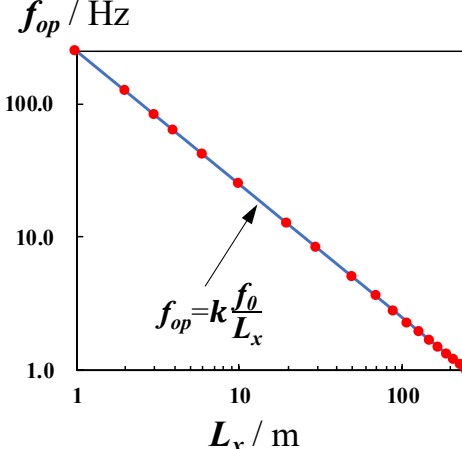

**Figure 8.** Relation between the optimal pulse frequency $f_{op}$ and pipe length $L_x$ when the wave velocity is 1000 m/s. The left drawing is plotted in linear coordinates and the right drawing is plotted in log–log coordinates. Here we take $k = 1$ as an example to show the curve.

### 3.2. Pressure Characteristic of Fluid When Considering Drag Effect of Wall

For the real flow in a pipe, the wall drag should be considered in most situations. It is vital to accurately calculate the one-way resistance or friction drag term. In this part, the drag effect of the wall is considered in the computation by adding the drag term including the friction drag coefficient. The friction drag coefficient $\lambda$ is closely related to the flow states. For the laminar flow, the friction drag coefficient is defined by $\lambda = 64/Re$, where $Re$ is defined by $Re = \rho u d / \mu$. For the turbulent flow, the friction drag coefficient is given by $1/\sqrt{\lambda} = 2log\left(Re\sqrt{\lambda}\right) - 0.8$, which is an implicit formula about $\lambda$. For the pipe flow, the classical Reynolds experiment confirmed that the laminar flow exists in the range of $Re < 2300$, and the turbulence exists approximately when $Re > 3000 \sim 4000$. In the range of $2300 < Re < 3000 \sim 4000$, the flow may be laminar or turbulent. This range of Reynolds numbers is also called the transition region. The friction drag coefficient of the transition region can be obtained by giving an average of friction drag coefficients in laminar and turbulent regions.

It is noteworthy that $Re$ depends on the pipe diameter and fluid velocity. Therefore, it is necessary to calculate $Re$ at every position of the pipe, which is the reason why $Re$ is also called the local Reynolds number. In fact, the calculation of $Re$ should be carried out at every time step due to the variation of fluid velocity. In PHF, our tests showed that the flow $Re$ varies in a wider range, $0 < Re < 106$ which covers the laminar flow, transition and turbulence. Hence, the friction drag coefficients mentioned above are all used in our calculation program. Essentially, the fluid velocity is related to the pressure difference of the fluid. Namely, the fluid velocity is affected by the pulse pressure applied at the inlet. Considering the reality of PHF, the average pulse pressure is set to 1 MPa at the inlet in the present cases. Some previous researchers [14,22] used similar pressure; for example, the pressure adopted by Zhai et al. was 0.5~2 MPa [14].

Another important thing to note is the computation of the friction drag coefficient for turbulence. It is given by $1/\sqrt{\lambda} = 2log\left(Re\sqrt{\lambda}\right) - 0.8$. This implicit formula cannot be directly solved to give the drag coefficient. To obtain the drag coefficient, we adopted the Newton iteration method to solve this implicit formula at every time step.

During the simulations, the initial fluid pressure is set to 1 MPa in the whole pipe, the average pressure of the sine pulse is also 1 MPa and the amplitude of the pulse is 0.1 MPa at the inlet. Without loss of generality, the pipe length is set to 10 m and the pulse frequency is set to 25 Hz. In the following discussion, we use the normalized pressure $P_n$ to describe the pressure characteristics of the fluid, where $P_n = 1$ represents the real fluid pressure $P = 1$ MPa.

Objectively, the flows in the pipe and fissure can be regarded as the one-dimensional flow during the PHF. From the viewpoint of one-dimensional flow, both the pipe and fissure can be simplified into a parallel channel, which is the same as the sectional drawing of the pipe shown in Figure 1. In our research, when $d > 10$ mm, the parallel channel (shown in Figure 1) represents the pipe. When $d < 10$ mm, the parallel channel represents the fissure. This division is more accordant with reality.

Results show that the pipe diameter or fissure aperture has a significant influence on fluid pressure, as shown in Figure 9. The peak pressure decreases when the pipe diameter and fissure aperture decrease. Comparing Figure 9a–c, it is found that the amplification effect of peak pressure is apparent when the pipe diameter is $d = 10$ mm, where the maximum peak pressure is enhanced by 100% (from $P_n = 1$ to $P_n = 2$) before the appearance of cavitation ($P_n \to 0$) (Figure 9c). When the crack aperture decreases to 1 mm, the amplification effect of pressure is also apparent, where the peak pressure is increased by 60% (Figure 9b). When the aperture further decreases to 0.1 mm, no amplification effect exists. Instead, the fluid pressure is suppressed by the narrow fissure, where the maximum peak pressure is only 1.002 at the endpoint (Figure 9a). Similar phenomena are observed at the midpoint shown in Figure 9. The detailed maximum peak pressures are listed in Table 1. These phenomena show that the fissure aperture does affect the magnification of peak pressure. Inside a too-narrow crack, the sine pulse at the inlet cannot provide an amplification effect.

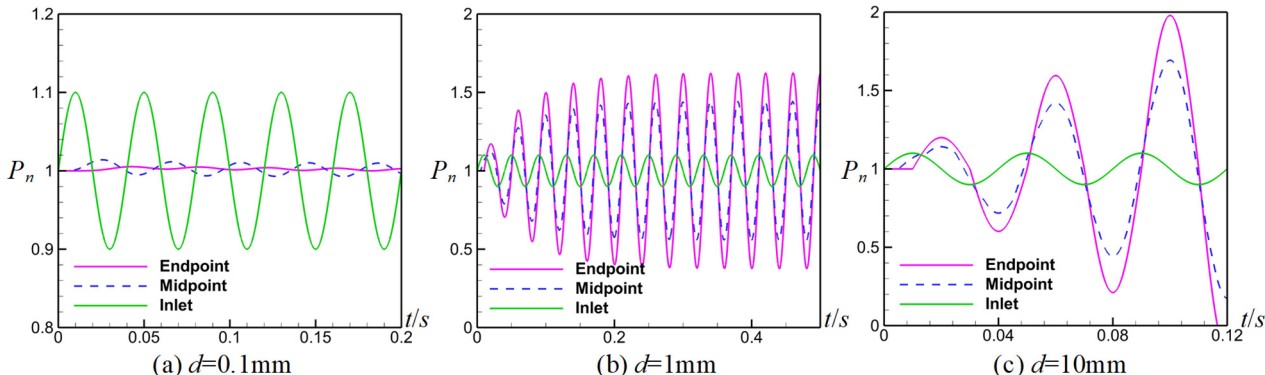

(a) $d$=0.1mm  (b) $d$=1mm  (c) $d$=10mm

**Figure 9.** Normalized pressure–time curves at inlet, middle point and endpoint, where $L_x = 10$ m and $f = 25$ Hz: (**a**) $d = 0.1$ mm, (**b**) $d = 1$ mm, (**c**) $d = 10$ mm.

**Table 1.** Inhibition or magnification effects of the pulse at midpoint and endpoint (partial data).

| $d$ (mm) | Peak Pressure at Midpoint | Effect | Peak Pressure at Endpoint | Effect |
|---|---|---|---|---|
| 0.1 | 1.009 | ✗ | 1.002 | ✗ |
| 0.2 | 1.029 | ✗ | 1.021 | ✗ |
| 0.3 | 1.048 | ✗ | 1.054 | ✗ |
| 0.4 | 1.076 | ✗ | 1.100 | = |
| 0.5 | 1.114 | ✔ | 1.155 | ✔ |
| 1 | 1.443 | ✔ | 1.625 | ✔ |
| 2 | 2.0 | ✔ | 2.0 | ✔ |
| 3 | 2.0 | ✔ | 2.0 | ✔ |
| 4 | 2.0 | ✔ | 2.0 | ✔ |
| 5 | 2.0 | ✔ | 2.0 | ✔ |

Note: ✗ represents inhibition, = represents no effect, ✔ represents magnification. The middle point is at $x = 5$ m, the endpoint is at $x = 10$ m.

Pay particular attention to the average pressure shown in Figure 9a. The average pressure at the inlet is the same as that at the endpoint. They both equal 1. The reason for this is given as follows: In our research, the right endpoint is a blind end, so there is no net inflow and net outflow of fluid. Hence, the average velocity is zero inside the

pipe or fissure. As a result, the average one-way friction drag is zero, causing the average pressure-drop to be zero from inlet to endpoint. Therefore, the average pressure at the endpoint equals that at the inlet. If the right endpoint is open and the fluid flows out, then the average pressure at the endpoint must be lower than that at the inlet due to the effect of wall friction drag.

Although the average velocity is zero in our research, the local velocity is not zero. During the calculation, we monitored the fluid velocity in the pipe and fissure. Here we take the midpoint as an example to explain. For the fissure of $d = 1$ mm, the fluid velocity at the midpoint varies periodically within the range of $\pm 0.43$ m/s, where the pressure fluctuation range is $1 \pm 0.443$ MPa. For the fissure of $d = 0.1$ mm, the fluid velocity at the midpoint varies periodically within the range of $\pm 0.0019$ m/s, where the pressure fluctuation range is $1 \pm 0.0087$ MPa. As a result, the average velocity is zero, the average pressure is 1 MPa and the average pressure-drop is 0, but the peak pressure is significantly suppressed in a narrower fissure such as $d = 0.1$ mm.

From Figure 10 and Table 1, we find that the peak pressure at the endpoint can be enhanced due to the effects of the inlet pulse. When $d > 0.9$ mm, the magnification effect is apparent, and the maximum peak pressure at the endpoint can be enhanced by more than 50%. When $d > 1.3$ mm, the maximum peak pressure at the endpoint can be increased by 100%, and the cavitation phenomena occur near the wave trough due to the appearance of a vacuum at the endpoint. However, in narrower fissures ($d < 0.9$ mm), the magnification is not apparent. Particularly, there is not any enhancement of peak pressure when $d < 0.4$ mm. Instead, inhibition effects are observed. It is noteworthy that the above phenomena and discussion are given under the condition that the pulse frequency is at the optimal frequency. In addition, the inlet average pressure is 1 MPa and the saturation vapor pressure is 0.0024 MPa when the water temperature is 20 °C [34]. Hence, the critical pressure of cavitation is $P_n = 0.0024$, which is very close to 0, in the $P_n$-$t$ plot shown in Figure 9c. The cavitation occurs when the minimum peak pressure is less than 0.0024 MPa ($P_n < 0.0024$). According to this criterion, the cavitation range is given in Figure 10. Our results indicate that for wider fissures and pipes ($d > 1.3$ mm), it is possible to further enhance the fluid pressure by inducing cavitation using the optimal pulse frequency. Usually, the local pressure can be enhanced to 50 MPa or higher during the collapse stage of cavitation bubbles. Therefore, it is vital to adopt the optimal pulse frequency to enhance the peak pressure of the fluid at the endpoint.

Comparing the midpoint and endpoint, it is found that the magnification effect is more apparent at the endpoint than at the middle point when $d > 0.4$ mm. The inhibition effect is also more apparent at the endpoint than at the middle point when $d < 0.4$ mm (shown in the local enlargement of Figure 10). Similar phenomena were reported by Zhai et al. (2015) [14].

We analyzed the maximum peak pressures of the endpoint at various pulse frequencies and crack apertures based on hundreds of cases. It is found that the optimal pulse frequency is constant regardless of whether the crack is narrow or wide, regardless of whether the diameter of the pipe is large or small. This optimal pulse frequency satisfies the frequency–length model proposed in the above section, namely $f = k f_0 / L_x$. This conclusion is right under the condition that the wave speed is constant. The explanation is given later.

This conclusion can be further explained by the property of wave propagation. The sine pulse applied at the inlet is the longitudinal wave, whose reflection and superposition are affected by the location of the blind end. Therefore, the optimal frequency is related to the length rather than the diameter of the pipe and the aperture of the fissure. The diameter and aperture affect the superposition strength of the pressure wave but do not affect the propagation speed of the pressure wave if the wave speed is constant. As a result, the peak pressure gradually decreases but the optimal frequency remains constant when the pipe diameter or fissure aperture is decreased under the condition of $a$ = constant.

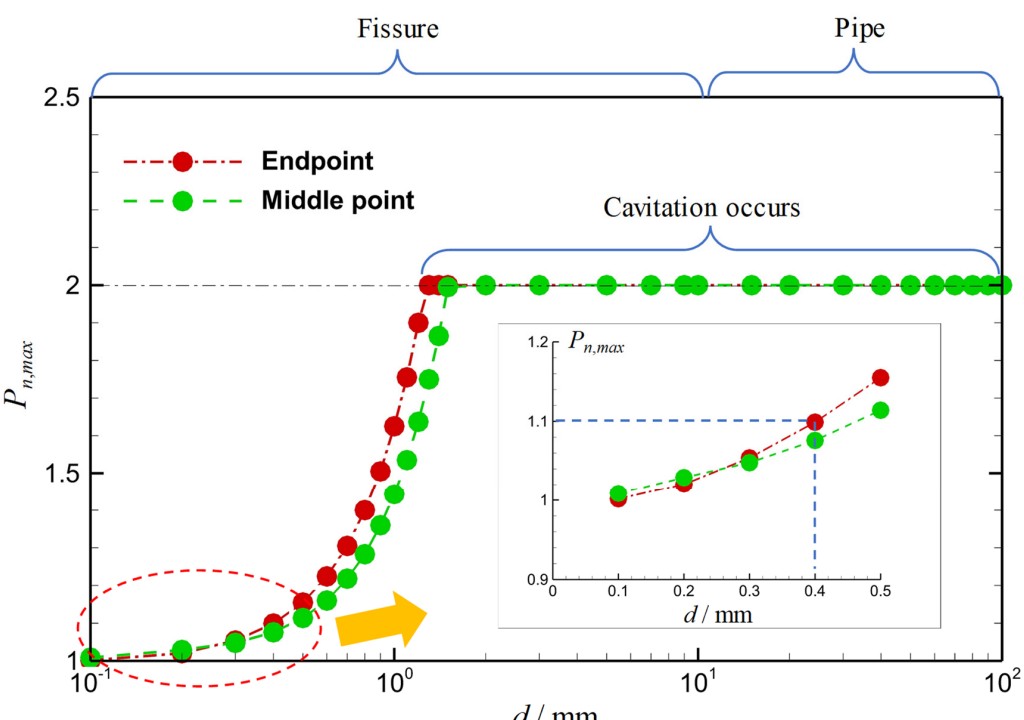

**Figure 10.** Distribution of normalized peak pressure $P_{n,max}$ where $L_x$ = 10 m and $f$ = 25 Hz. When $d \geq 10$ mm, $d$ represents the diameter of the pipe. When $d < 10$ mm, $d$ represents the aperture of the fissure.

It is important to note that, in fact, the wave speed is related to the pipe diameter, which will be explained in Section 3.4. The above discussion suggests that the pipe diameter does not affect the wave speed. This discussion is right only under the condition of $a$=constant. This condition is reasonable because there are some situations where the wave speed remains constant when the pipe diameter is increased or decreased. A detailed explanation is also given in Section 3.4.

*3.3. Effect of pulse Amplitude on the Peak Pressure Characteristic of Fluid*

The amplitude is another important parameter affecting the pressure distribution. Three kinds of amplitudes are used to analyze their influence on the peak pressure at the endpoint: $A$ = 0.1, 0.2 and 0.3. Because the normalized pressure is a dimensionless quantity, the amplitude is nondimensional. Without loss of generality, we choose a smooth pipe whose length is 10 m. The wave speed is set to 1000 m/s. The detailed results are given as follows:

At lower frequency $f$ = 1 Hz (shown in Figure 11a–c), it is found that the pressure at the endpoint is the same as that at the inlet when the frequency is 1 Hz. The peak pressure increases from 1.1 to 1.3 when the amplitude varies from 0.1 to 0.3. The magnification of amplitude at the endpoint is the same as that at the inlet. A similar phenomenon is observed at higher frequency $f$ = 100 Hz (shown in Figure 11g–i). When the amplitude varies from 0.1 to 0.3, the maximum peak pressure increases from 1.2 to 1.6; namely, the maximum peak pressure is enhanced from 20% to 60%. At the optimal pulse frequency $f$ = 25 Hz (shown in Figure 11d–f), the cavitation phenomena exist because there is extremely low pressure. However, the time of reaching cavitation decreases with the increase in amplitude. When $A$ = 0.1, the zero pressure appears at $t$ = 0.116 s (near the third wave trough). When A = 0.2, the zero pressure appears at $t$ = 0.074 s (near the second wave trough). When A = 0.3, the zero pressure appears at $t$ = 0.036 s (near the first wave trough). These phenomena indicate that the magnification effect can be strengthened by increasing the pulse amplitude.

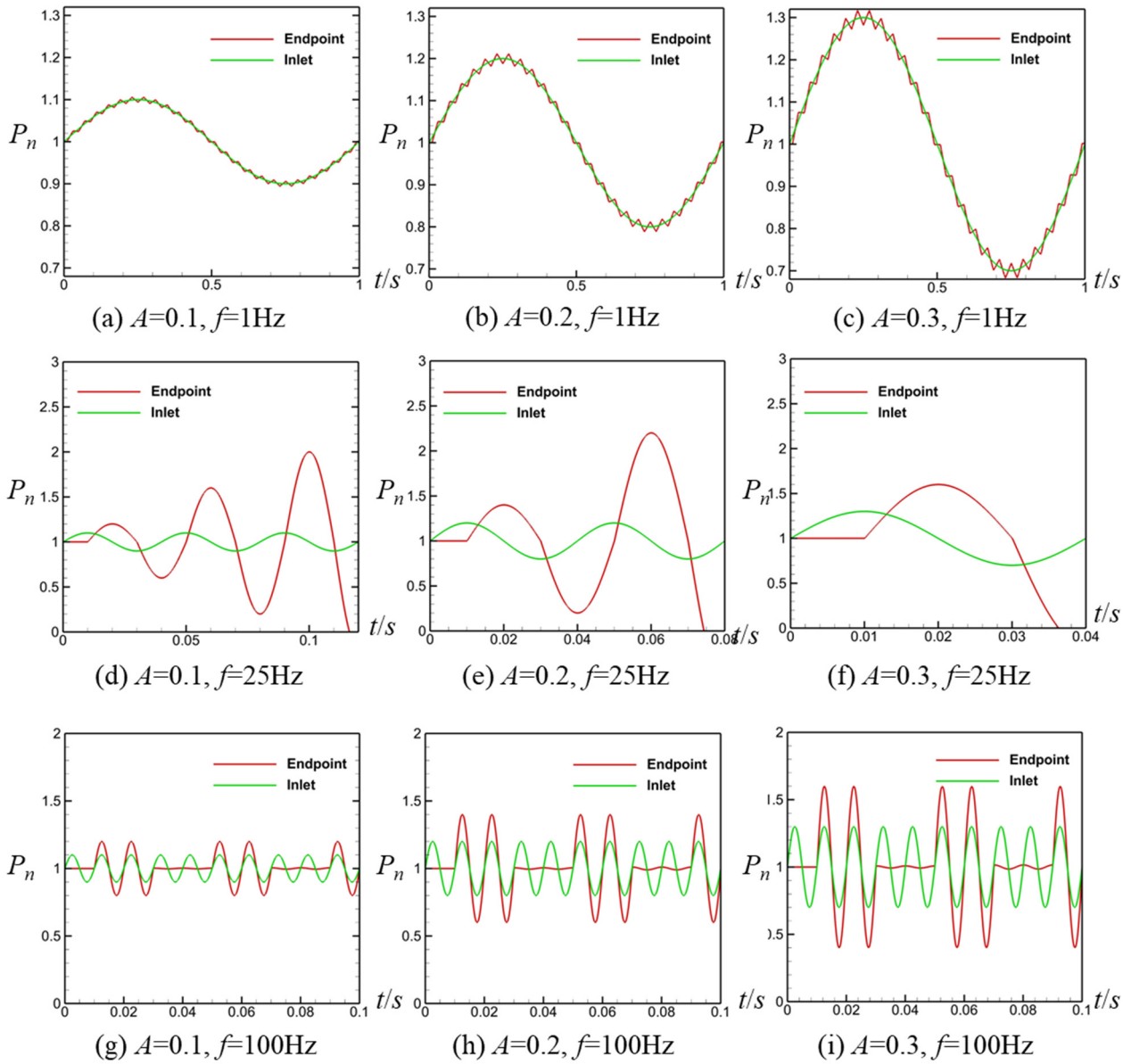

**Figure 11.** Normalized pressure–time curves where $L_x$ = 10 m and $a$ = 1000 m/s: (**a**) $A$ = 0.1, $f$ = 1 Hz; (**b**) $A$ = 0.2, $f$ = 1 Hz; (**c**) $A$ = 0.3, $f$ = 1 Hz; (**d**) $A$ = 0.1, $f$ = 25 Hz; (**e**) $A$ = 0.2, $f$ = 25 Hz; (**f**) $A$ = 0.3, $f$ = 25 Hz; (**g**) $A$ = 0.1, $f$ = 100 Hz; (**h**) $A$=0.2, $f$ = 100 Hz; (**i**) $A$ = 0.3, $f$ = 100 Hz.

The amplitude of the sine pulse affects the peak pressure of fluid inside the pipe. However, the optimal frequency is not influenced by the amplitude of the sine pulse applied at the inlet. The essential reason is that the pulse pressure propagates in the form of a longitudinal wave rather than a transverse wave. The traveling direction of the longitudinal wave is parallel to the pipe wall and the wave speed is constant. Hence, the optimal pulse frequency does not depend on the initial amplitude of the pulse. In conclusion, the optimal pulse frequency is independent of the pulse amplitude applied at the inlet of the pipe.

To improve the fracturing effect of PHF, it is necessary to enhance the pulse pressure of the hydraulic fluid inside the pipe. Based on the above research, we propose the following fracturing strategies: To give a higher pulse pressure, two effective ways can be adopted. The first one is to choose the optimal frequency at the inlet. The second one is to increase the pulse amplitude at the inlet if possible. The best method is to combine both ways to enhance the peak pressure of the hydraulic fluid.

### 3.4. Influence of Wave Speed on the Optimal Frequency

In the above discussion, it is assumed that, in water, the pressure wave travels at speed of 1000 m/s along the pipe direction. However, in a real project, the wave speed may change around 1000 m/s. The pressure characteristic would change in space and time dimensions. In addition, the optical pulse frequency must change due to the alteration of wave speed. Therefore, it is essential to discuss the effects of wave speed on the pressure distribution and optimal pulse frequency. Considering that the drag effect does not influence the optimal frequency confirmed above, we discuss the wave speed effects only in a smooth pipe in this part.

The traveling speed of a pressure wave is related to the physical properties of fluid, the material of the pipe and rock, and the geometric features of the pipe. The definition of wave speed is $a = c/\sqrt{(1 + 4E_w/(d \cdot K)}$, where $c$ is the sound velocity in water and equals 1476 m/s at 20 °C; $E_w$ is the elastic modulus of water and equals $2.18 \times 10^3$ MPa; and $K$ is the resistance coefficient, whose unit is Pa/m. In fact, the resistance coefficient $K$ is a complex function of the pipe elastic modulus, pipe diameter, pipe wall thickness, rock elastic modulus, Poisson ratio, and so on. The function form of the resistance coefficient depends on the pipe installation mode (e.g., exposed pipe, deep-buried pipe, with stiffener or not). Taking an exposed steel pipe as an example, the definition of $K$ is $K = 4E_s \delta/d^2$. Then, the wave speed of pressure is $a = c/\sqrt{(1 + (E_w \cdot d)/(E_s \cdot \delta))}$, where $E_s$ is the elastic modulus of steel and equals $2.06 \times 10^5$ MPa, $d$ is the diameter of pipe and $\delta$ is the wall thickness of the pipe. For a deep-buried pipe, the definition of $K$ is more complex.

Objectively, the resistance coefficient and sound velocity depend on many other variables, such as material, temperature and fluid properties. Hence, it is senseless and tedious to analyze every influencing factor when discussing the influence of wave speed on the optimal pulse frequency. Instead, it is practicable to give a range of wave speeds and then discuss the effects of wave speed in this range because the wave speed always falls within a range no matter how the related influence factors are adjusted and combined. In view of these reasons, we discuss the effect of wave speed in a practical range from $a = 500$ m/s to $a = 1500$ m/s.

As an example, the pipe length is set to 25 m. The sine pulse is applied at the inlet with a given frequency. The pulse amplitude is 0.1 normalized by the average pressure of the sine pulse at the inlet. The normalized pressure is recorded at the endpoint during the PHF. For every wave speed, the normalized pressure–time curve is obtained from low frequency to high frequency with a small frequency interval. Partial results are shown in Figure 12.

Under the condition $a = 800$ m/s, Figure 12a shows that the peak pressure at the endpoint gradually increases with time when the frequency is 8 Hz. There is an approximately linear correlation between the peak pressure and the time. The maximum peak pressure is 2 at $t = 0.33$ s (Figure 12a) and then the pressure drops to zero, causing the appearance of cavitation. At $f = 10$ Hz (Figure 12b), the maximum peak pressure is near 1.5, which is lower than that at $f = 8$ Hz. When further increasing the frequency to 12 Hz (Figure 12c), the maximum peak pressure is only 1.3, indicating that the magnification is not apparent as those at $f = 8$ Hz and $f = 10$ Hz. By comparing a great number of cases with different frequencies, it is found that the optimal frequency is 8 Hz when the wave speed is 800 m/s.

Under the condition $a = 1000$ m/s, Figure 12d,f shows that the peak pressures are near 1.6 when the pulse frequency is 8 Hz and 12 Hz. In contrast, the peak pressure exhibits an approximately linear increase with time when the pulse frequency is 10 Hz, and the peak pressure increases to 2 at $t = 0.29$ s (Figure 12e). The magnification at $f = 10$ Hz is significantly larger than those at other pulse frequencies such as $f = 8$ Hz and $f = 12$ Hz. At last, it is found that the optimal frequency is 10 Hz when the wave speed is 1000 m/s.

Under the condition $a = 1200$ m/s, Figure 12g,h show that the peak pressure is near 1.3 and 1.7 when the pulse frequency is 8 Hz and 10 Hz, respectively. In contrast, the peak pressure is 2 at $t = 0.24$ s (Figure 12i) when the pulse frequency is 12 Hz. The magnification at $f = 12$ Hz is significantly larger than those at other pulse frequencies such as $f = 8$ Hz and

$f = 10$ Hz. At last, $f = 12$ Hz is identified as the optimal pulse frequency when the wave speed is 1200 m/s.

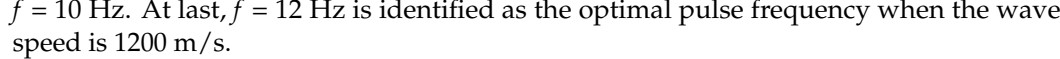

**Figure 12.** Normalized pressure–time curves where the pipe length is $L_x = 25$ m. The upper part (**a–c**) is obtained when the wave speed is 800 m/s, the middle part (**d–f**) is obtained when the wave speed is 1000 m/s, the lower part (**g–i**) is obtained when the wave speed is 1200 m/s.

These phenomena show that the optimal frequency $f_{op}$ depends on the wave speed $a$. To completely clarify the influence of wave speed, it is necessary to research the optimal frequency inside pipes with different lengths. Considering the influence of pipe length and wave speed, we conducted hundreds of numerical experiments. For the first time, it is found that there is a quantitative relationship between the optimal pulse frequency $f_{op}$, pipe length $L_x$ and wave speed $a$. The optimal frequency is closely related to the ratio of wave speed $a$ and pipe length $L_x$. The quantitative relationship is $f_{op} = ka/(4L_x)$. The parameter $k$ is any positive odd number and reflects the periodicity of a wave. The detailed results are

shown in Figure 13. Our results confirm that this quantitative formula is right for any pipe length and wave speed. The quantitative formula shows that the optimal pulse frequency is proportional to wave speed and inversely proportional to pipe length.

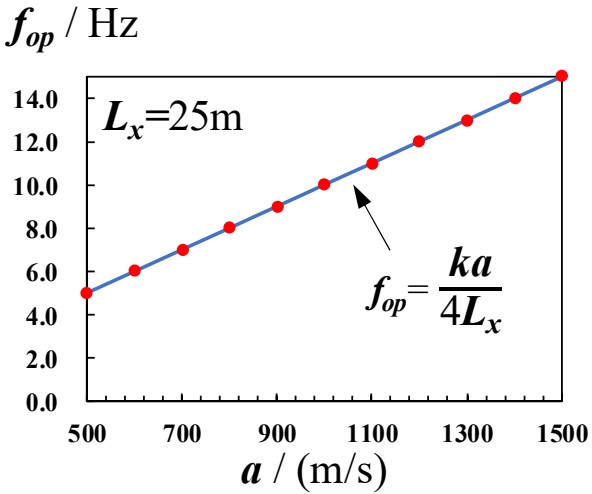
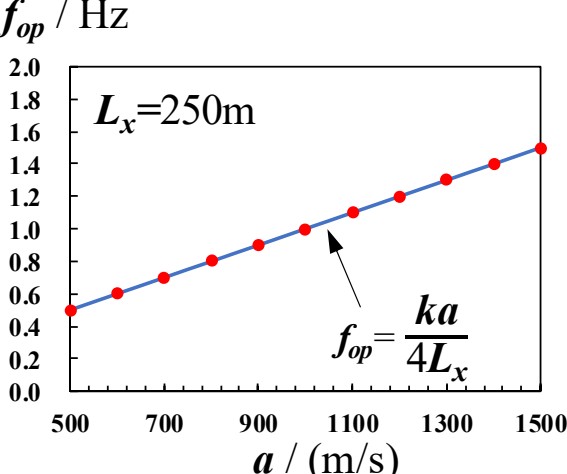

**Figure 13.** Relationship between the optimal pulse frequency and wave speed with different pipe lengths. The left data points are obtained when the pipe length is 25 m, and the right data points are obtained when the pipe length is 250 m. Here $k = 1$ is taken as an example.

According to the definition of wave speed, the wave speed is related to the pipe diameter. Here, we take an exposed steel pipe as an example; the wave speed is $a = c/\sqrt{(1 + (E_w \cdot d)/(E_s \cdot \delta))}$. There are some situations where the optimal pulse frequency is independent of the pipe diameter if the ratio $d/\delta$ is constant. This is because no matter how the pipe diameter is increased or decreased, the wave speed $a$ is constant if $d/\delta$ = constant. Therefore, the optimal frequency directly depends on the wave speed $a$ and pipe length $L_x$ rather than the pipe diameter or fissure aperture $d$. In conclusion, the optimal pulse frequency is identified by $f_{op} = ka/(4L_x)$, which is proposed by us for the first time.

### 3.5. Discussion on the Supercharging Mechanism

The supercharging phenomena have been shown in the above sections. However, the supercharging mechanism has not been revealed and discussed in detail. To clarify the inner mechanism of supercharging, we discuss the supercharging process and analyze the supercharging principle by studying the transient evolution characteristics of pressure and velocity.

It is not necessary to discuss every case with different pipe lengths, wave speeds and so on. In contrast, it is meaningful to focus on a particular case to research the supercharging process in the time and space dimensions. Without loss of generality, we choose a case where the pipe length is 250 m, the wave speed is 1000 m/s and the amplitude of the sine pulse is 0.1 $P_n$. In addition, the drag effect of the wall is not considered temporarily, which does not influence the optimal frequency if $a$ = constant, as confirmed in the above section.

To recreate the supercharging process, according to the formula $f_{op} = ka/(4L_x)$, the pulse frequency is set to 1 Hz for the present case where $L_x = 250$ m and $a = 1000$ m/s. The pulse period is $T = 1/f = 1$ s. The other conditions are the same as those mentioned in the above section, including the initial and boundary conditions.

During the computation, the initial fluid pressure is set to 1 MPa in the whole pipe, the average pressure of the sine pulse is also 1 MPa and the amplitude of the pulse is 0.1 MPa at the inlet. In the following discussion, we used the normalized pressure $P_n$ to describe the pressure characteristics of the fluid, where $P_n = 1$ represents the real fluid pressure $P = 1$ MPa.

At $t = 0$ s, the normalized pressure is 1 in the whole pipe. At $t = 0.25$ s, the pressure increases to 1.1 at inlet ($x = 0$ m). Here, 0.25 s is a 1/4 pulse period. According to the wave speed $a = 1000$ m/s, the pressure wave travels to the endpoint ($x = 250$ m) at $t = 0.25$ s. At this moment, a 1/4 sine wave forms, as shown in Figure 14a. Then, during the second 1/4 period (0.25 s~0.5 s), the inlet pressure gradually decreases to 1, and the endpoint pressure gradually increases to 1.2, as shown in Figure 14b. It is easy to understand the decrease in inlet pressure because it is controlled by the inlet boundary condition. The pressure $P_n = 1.2$ at the endpoint indicates that there is a supercharging process, causing the magnification of pressure at the endpoint. At $t = 1.5$ s, the magnification is more apparent, and the pressure reaches 1.6 at the endpoint (Figure 14f). At $t = 2.5$ s, the pressure further increases to 2 at the endpoint (Figure 14j). The detailed temporal evolution of pressure is given in Figure 14k.

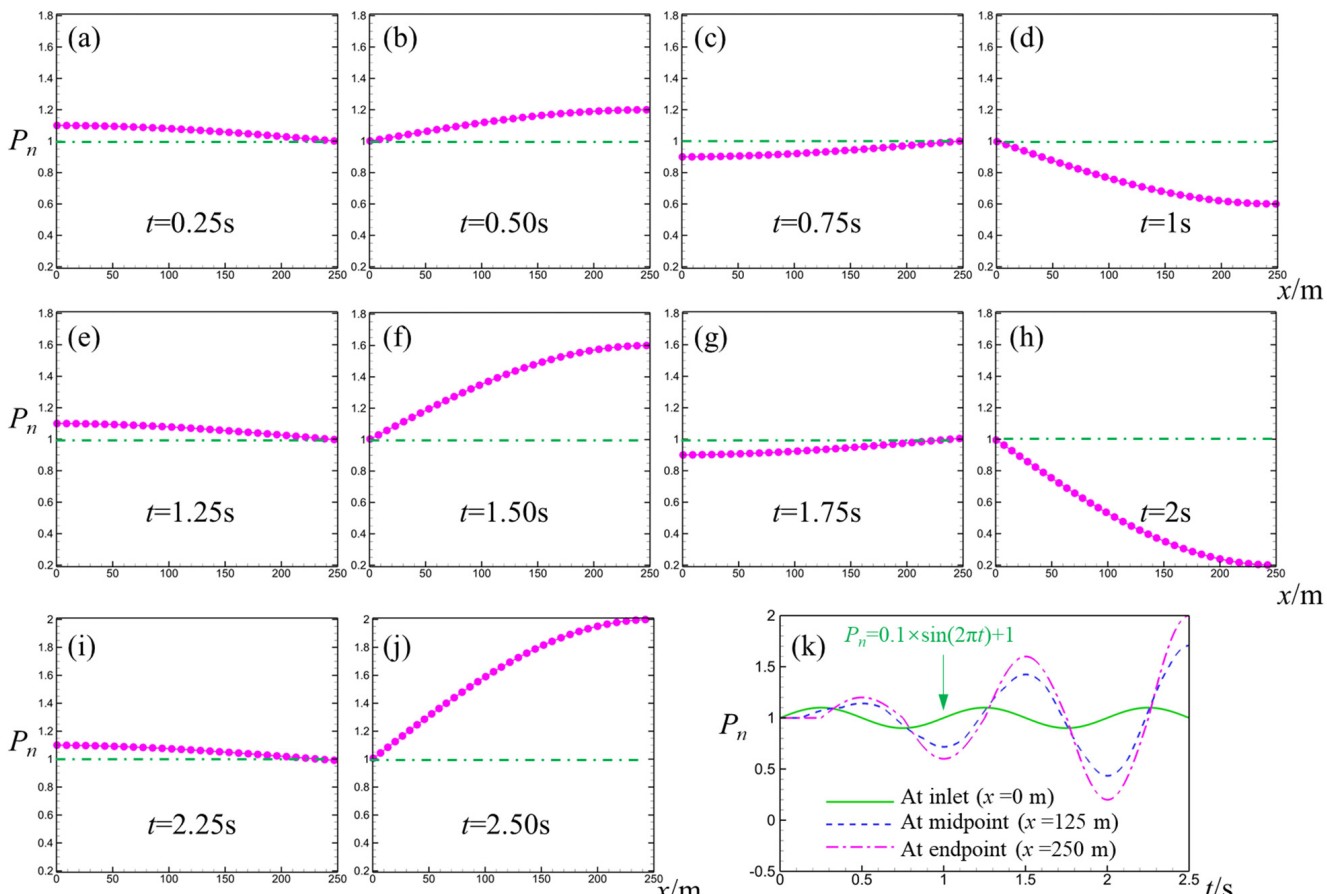

**Figure 14.** Normalized pressure–position curves during 2.5 pulse periods where $L_x = 250$ m and $f = 1$ Hz. (**a–j**) $t = 0.25$–2.5 s, (**k**) Normalized pressure–time curves during 2.5 pulse periods at inlet, midpoint and endpoint.

From Figure 14k, it can be seen that the inlet pressure (at $x = 0$ m) periodically changes in a sine manner, and its peak pressure is constant, equaling 1.1. However, the peak pressure at the endpoint ($x = 250$ m) periodically increases because of the periodic magnification of amplitude at the endpoint. The peak pressure increases from 1.2 at $t = 0.5$ s to 1.6 at $t = 1.5$ s and then further increases to 2 at $t = 2.5$ s. Similar phenomena are observed at the midpoint ($x = 125$ m), but the magnification effect is weaker than that at the endpoint.

To further explain the magnification phenomenon of amplitude at the endpoint, we focus on the second 1/4 pulse period 0.25 s $\leq t \leq$ 0.5 s. More detailed evolution processes are computed and extracted, as shown in Figure 15, including the pressure contours and the pressure–time and velocity–time curves. At $t = 0.25$ s, the pressure and velocity follow

a 1/4 sine distribution in the spatial direction (pipe-axis direction). The fluid velocity is 0.1 m/s at the inlet ($x$ = 0 m). The fluid velocity in the pipe is larger than 0, indicating that there is a dynamic impact load in the $x$ direction. Figure 15 shows that the endpoint pressure gradually increases from $P_n$ = 1 to $P_n$ = 1.2 when the time increases from $t$ = 0.25 s to $t$ = 0.5 s. There are two main reasons causing the increase in pressure at the endpoint. The first reason is the self-reflection of the pressure wave, which provides the basic part of pressure enhancement. The second reason is the transformation from dynamic energy to pressure energy, which provides the additional part of pressure enhancement. Continuous impact loads are injected at the inlet, and these dynamic energies are periodically transformed into pressure at the endpoint. Therefore, the peak pressure is periodically enhanced at the endpoint.

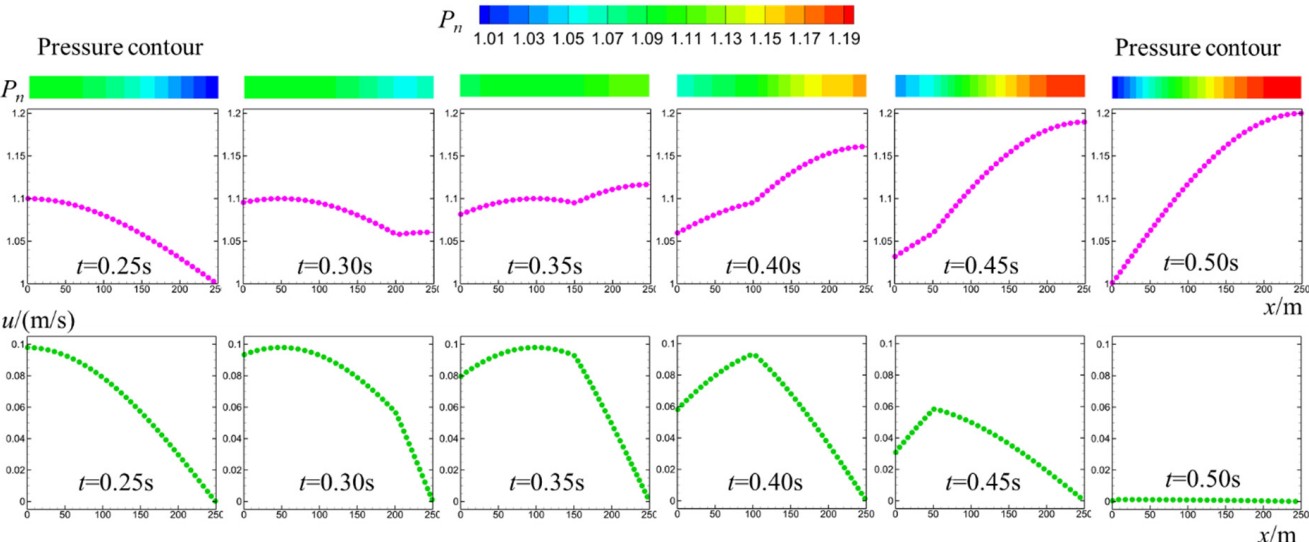

**Figure 15.** Normalized pressure–position $P_n$-$x$ curves (upper part) and velocity–position $u$-$x$ curves (lower part) during 0.25 pulse-periods from $t$ = 0.25 s to $t$ = 0.5 s, where $L_x$ = 250 m and $f$ = 1 Hz. The velocity is not zero at $t$ = 0.25 s (shown in the lower left).

The supercharging from dynamic energy can be further confirmed from an example. Here, it is assumed that all the conditions are the same as in the above case except the dynamic energy at $t$ = 0.25 s. In other words, there is no dynamic energy in the pipe at $t$ = 0.25 s; i.e., the fluid velocity is 0 at this moment. The evolutions of pressure and velocity are shown in Figure 16. It can be seen that the endpoint pressure gradually increases, but it only increases to $P_n$ = 1.1 at $t$ = 0.5 s, which is less than the pressure $P_n$ = 1.2 shown in Figure 15 at $t$ = 0.5 s. The key reason is that there is no additional enhancement of pressure because of the absence of initial dynamic energy. Comparing Figures 15 and 16, it is confirmed that the dynamic energy applied at the inlet is the necessary condition for enhancing the peak pressure at the endpoint. Objectively, the total energy is conserved at $t$ = 0.25 s and $t$ = 0.5 s. The additional increase in pressure energy at $t$ = 0.5 s comes from the initial dynamic energy of the fluid at $t$ = 0.25.

In fact, the essential reason for supercharging is the reflection of pressure waves and the transformation of dynamic energy. The supercharging effect is significant only when the inlet sine pulse is applied at the optimal frequency, at which the reflection and transformation resonate.

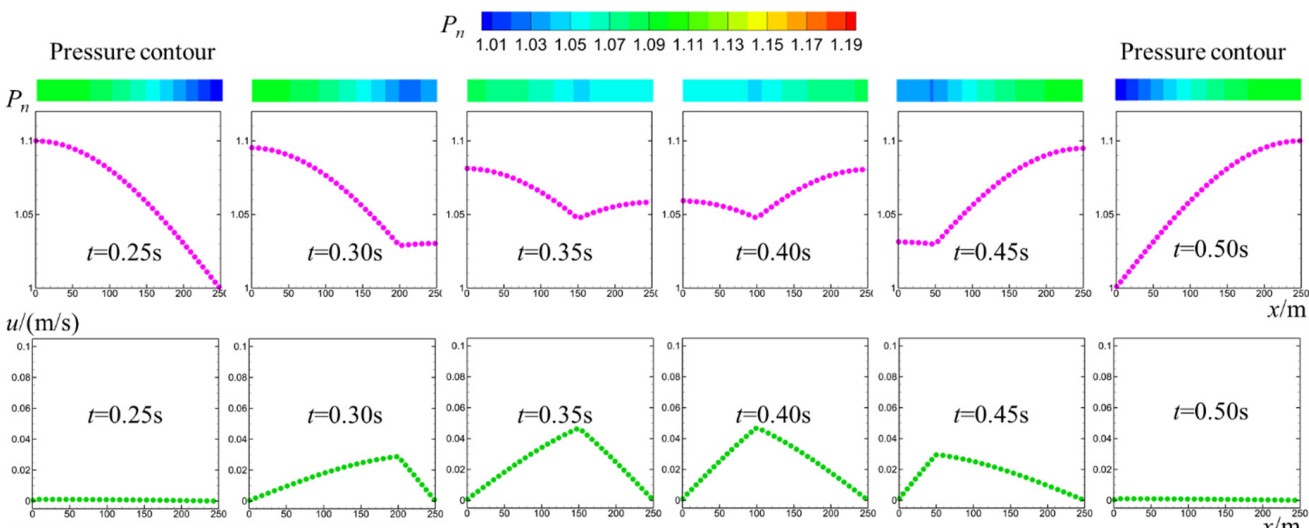

**Figure 16.** Normalized pressure–position $P_n$-$x$ curves (upper part) and velocity–position $u$-$x$ curves (lower part) during 0.25 pulse-periods from $t$ = 0.25 s to $t$ = 0.5 s, where $L_x$ = 250 m and $f$ = 1 Hz. The velocity is zero at $t$ = 0.25 s (shown in the lower left). The inlet average pressure is $p_{av}$ = 1 MPa.

In this paper, we discussed the supercharging phenomena of fluid pressure inside the pipe and revealed the inner mechanism of supercharging. In addition, we found the optimal pulse frequency, and we also gave a quantitative formula to identify the optimal frequency for the first time. This new universal formula is emphasized as follows:

$$f_{op} = \frac{ka}{4L_x} \tag{15}$$

This formula or model shows that the optimal pulse frequency $f_{op}$ is proportional to wave speed $a$ and inversely proportional to pipe length $L_x$, which has been strictly validated by the numerical experiments and theory analysis above. The parameter $k$ is any positive odd number, and it represents a family of frequencies due to the periodicity of the sine wave.

It is confirmed that the maximum peak pressure at the endpoint can be enhanced by 100% or more, which is larger than the traditional results. It is revealed that cavitation phenomena exist when the optimal frequency is applied. In this situation, the magnification of peak pressure is far larger than 2 times. Therefore, the present frequency-control method has obvious advantages. Detailed comparisons are given in Table 2. The present method has great potential in PHF projects due to its ability to remarkably enhance the peak pressure of a fluid.

**Table 2.** Comparison between the present sine PHF method and traditional PHF method.

| Method | How to Choose the Optimal Frequency | Magnification of Peak Pressure | Magnification Effect |
|---|---|---|---|
| Traditional method | No criterion | <20% | Inapparent |
| Present method | $f_{op} = \frac{ka}{4L_x}$ | 100% or more | Apparent |

Although we have proposed a universal model, there is still some work to do. In the present research, we mainly focused on the initial stage of PHF before the breakdown of rock. In this stage, the right endpoint of the pipe can be regarded as a blind end, which is vital for the present research. However, when the rock breaks and fractures appear, the endpoint of the pipe cannot be assumed as an ideal blind end. In this situation, the optimal pulse frequency may slightly change. These more complex cases need further studies.

Although we discussed the drag effects of the parallel fissure (similar to pipe) partially in Section 3.2, the present discussions about fissures only suit a main fracture that has the characteristic of approximately parallel walls without any proppant [35]. In a complex fracture network, the small-size fractures are tortuous, so the propagation regularity of pressure waves needs further studies in fracture networks. Nonetheless, the present work lays the foundation for these challenging problems. In addition, the fluid pressure affects the rupture of the rock, further influencing the number of fissures/fractures. Usually, the number of fissures increases with the pulse peak pressure. In addition, the fissure number is affected by other parameters, as pointed out by Mukhtar et al. (2022) [36]. The influence of the PHF parameters on the fissure number needs further study.

Another challenge is that of realizing the frequency control in technology based on the present theory, method and model. In fact, the supercharging process is closely related to the injection frequency of fluid controlled by the pulse pump. Therefore, it is vital to accurately control the pulse pump. It is necessary and valuable to achieve precise control of frequency by researching pumping equipment and control processes. Still, the present research and findings provide a foundation for these challenges.

The present method shows a huge application potential. In the next step, we plan to design a pulse pump to achieve the pulse supercharging process. There is much work to do, including the design of the pulse pump. In fact, the pulse pump is not a traditional water pump. The pulse pump in our study has particular flow rate demands. We are attempting to design a pulse pump that can be accurately controlled by flow rate.

## 4. Conclusions

In this paper, we studied the effects of the sine pulse on the peak pressure of fluid inside a pipe and fissure using a theoretical model and numerical simulations. The simulations were conducted by developing a computation model and writing a calculation program, and the results were strictly validated by experiments. The influence of sine pulse frequency was researched in a wide frequency range, and the optical frequency was found at last. The pressure distribution properties were analyzed in the time and space dimensions. The present research reveals a new phenomenon and law of peak pressure. These important findings and significant conclusions are as follows:

(1) It is first found that the peak pressure of fluid can be significantly enhanced at the pipe or fissure end face if the sine pulse is continually applied at the inlet. The peak pressure at the end face can be enhanced by more than 100% relative to the inlet average pressure. This new finding is remarkably different from the traditional viewpoint and conclusions, which suggest that the peak pressure can only be enhanced by less than about 20%.

(2) It is found that the optimal frequency is related to the pipe (or fissure) length and wave speed. It is confirmed that there is a quantitative relationship between the optimal pulse frequency, pipe length and wave speed. The quantitative relationship is $f_{op} = ka/(4L_x)$. This universal formula shows that the optimal pulse frequency is proportional to the wave speed and inversely proportional to the pipe length or fissure departure.

(3) The pulse amplitude affects the peak pressure of fluid but does not influence the optimal frequency. It is confirmed that the enhancement of peak pressure at the endpoint is proportional to the magnification of amplitude at the inlet before the appearance of cavitation.

(4) The peak pressure of the fluid at the endpoint is influenced by the pipe diameter or fissure aperture. When $d \geq 1.3$ mm, the maximum peak pressure is enhanced by 100% or more and the cavitation phenomena exist at the endpoint. However, in narrower fissures ($d < 0.9$ mm), the magnification is not apparent. Instead, inhibition effects are observed when $d < 0.4$ mm due to the influence of friction drag. These conclusions are obtained under the condition of optimal pulse frequency.

(5) This is the first time that the supercharging mechanism is revealed in the sine PHF. The fluid pressure at the end face is significantly enhanced due to two main reasons. The first reason is the self-reflection of the pressure wave. The second reason is the transformation of dynamic energy. Only at the optimal pulse frequency, resonance occurs

between the self-reflection of the pressure wave and the transformation of dynamic energy, causing the apparent supercharging of the fluid at the endpoint.

**Author Contributions:** Conceptualization, investigation, writing and project administration H.L.; Investigation, supervision, project administration B.H.; Investigation, supervision; project administration H.X. All authors have read and agreed to the published version of the manuscript.

**Funding:** This work is supported by the Natural Science Foundation of Jiangsu province of China (Grant No. BK20221123) and the Fundamental Research Funds for the Central Universities (Grant No. 2022QN1018 and 102522170). The research is also financially supported by the State Key Laboratory of Coal Resources and Safe Mining, China University of Mining and Technology (Grant No: SKLCRSM22X003).

**Institutional Review Board Statement:** Not applicable.

**Informed Consent Statement:** Not applicable.

**Data Availability Statement:** The data that support the findings of this study are available from the author upon reasonable request.

**Acknowledgments:** Thank my tutor Xu for his guidance in CFD.

**Conflicts of Interest:** The authors declare no conflict of interest.

## Nomenclature

| | | | |
|---|---|---|---|
| $a$ | Wave speed, m/s | $t_p$ | Duration time of maximum peak pressure, s |
| $A$ | Pipe cross-section area, m$^2$ | $T$ | Pulse period, s |
| $A_p$ | Pulse amplitude, 1 | $u$ | Fluid velocity, m/s |
| $c$ | Sound velocity, m/s | $x$ | Streamwise coordinate, m |
| $d$ | Pipe diameter, m | $\Delta p$ | Pressure variation, Pa |
| $E_s$ | Elastic modulus of steel, MPa | $\Delta t$ | Time step size, s |
| $E_w$ | Elastic modulus of water, MPa | $\Delta u$ | Velocity variation, m/s |
| $f$ | Pulse frequency, Hz | $\Delta x$ | Grid element size, m |
| $f_{op}$ | Optimal pulse frequency, Hz | $\delta$ | Pipe wall thickness, m |
| $F$ | Force, N | $\lambda$ | Friction drag coefficient, 1 |
| $g$ | Gravity acceleration, m/s$^2$ | $\mu$ | Molecular viscosity of fluid, Pa•s |
| $h$ | Water head, m | $\rho$ | Fluid density, kg/m$^3$ |
| $H_w$ | Constant water head at inlet, m | **Superscript** | |
| $k$ | Positive odd number, 1 | $t$ | Value at time $t$ |
| $K$ | Resistance coefficient, Pa/m | $-$ | Predicted value |
| $L_x$ | Pipe length, m | **Subscript** | |
| $m$ | Mass of fluid element, kg | 0 | Location at inlet or left element |
| $p$ | Fluid pressure, Pa | 1 | Location of the first grid point |
| $P_n$ | Normalized pressure, 1 | nx | Location next to the endpoint |
| $P_{n,max}$ | Maximum peak pressure, 1 | nx+1 | Location at endpoint |
| $\mathbf{Re}$ | Reynolds number, 1 | av | Average value |
| $t$ | Time, s | $i$ | The ith index |

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
