# Peer review of "The Optimal Sine Pulse Frequency of Pulse Hydraulic Fracturing for Reservoir Stimulation"

_water, doi:10.3390/w14193189_

Round 1

Reviewer 1 Report

The paper presents an interesting numerical study on the optimal pulse frequency of pulse hydraulic fracturing for reservoir stimulation. The work is detailed. But the author should address the following before the manuscript can be accepted.

1. Page 1: Line 5 in the first paragraph under the Introduction section: What is the meaning of PH?

2. Page 2: 5th line in the last paragraph: In our… should be reworded without the personal pronoun “our”

3. Page 2: 10th line in the last paragraph: “had roof”… should be rewritten as “hard roof”

4. Page 3: 3rd line in the second paragraph: “Their study gave meaningful finding” should be deleted. Instead, the finding should just be stated

5. Page 3: 5th line in the second paragraph: “Their researches … ” should be rewritten as “Their study …”

6. Page 3: 2nd to the last line in the second paragraph: “Their researches … ” should be rewritten as “Their study …”

7. Page 3: 1st line in the third paragraph: The comma in “The first way is that, the fluid pressure … ” should be deleted. The same comment applies to the 3rd line in the same paragraph that reads “The second way is that, the fluid pressure …”

8. Page 3: Last paragraph: The findings of the study should not be mentioned there. This last paragraph should focus on highlighting the aim and objectives of the study and summarizing how the work addresses the question raised by the research.

9. Page 4: 5th line under Section 2: “… include the continue …” should be corrected. What is "continue equation"? The same correction should be done on the remaining parts of the paper where this wrong nomenclature is mentioned.

10. Page 5: Last line of the first paragraph: “…. in the discrete form” should be rewritten as “…. in the discrete form given by Eq. (4).”

11. Page 6: 2nd line in the paragraph above equation (10): What does the author mean by “continue sine pulse… ”?

12. Page 7: The author mentioned in the first paragraph of Page 7 that the work focus on the loading stage

of pulse pressure before the breakdown of rock. Comments should be added to justify this choice and also to add some remarks about what the limitation and applicability of this approach are.

13. Page 7: 6th paragraph: the word “we” should be replaced with something more appropriate such as “this study considers …” or something similar.

14. The study should comment on the implication of such PHF simulation and the pressure behavior on the number of fissures/fractures generated in real life (in the field). Single versus multiple fracture simulation in non-PHF studies showed some remarkable differences in the treatment parameters. The author should add to the manuscript, the following recent paper that discussed this fact on conventional HF and the manuscript should make suggestions as to how these phenomena differ in the PHF.

• Mukhtar, F. M., Shauer, N., & Duarte, C. A. (2022). Propagation mechanisms and parametric influence in multiple interacting hydraulic fractures: A 3-DG/XFEM hydro-mechanical modeling. International Journal for Numerical and Analytical Methods in Geomechanics.

15. It would benefit the readers if the author can add a new section dedicated to the practical applicability of the simulation so conducted and how its findings can be used as a valuable guide for the PHF design (as previously claimed in the introduction section of the manuscript).

16. The English language needs a review by a competent language editor.

Author Response

  1. Page 1: Line 5 in the first paragraph under the Introduction section: What is the meaning of PH?

Reply: Thank you for the careful review. HF was mistaken as PH, and we correct it in the first paragraph. The meaning of HF is given in the in the first paragraph.

  1. Page 2: 5th line in the last paragraph: In our… should be reworded without the personal pronoun “our”

Reply: Thank you for the careful review. The personal pronoun “our” was replaced by “the”.

  1. Page 2: 10th line in the last paragraph: “had roof”… should be rewritten as “hard roof”

Reply: Thank you for the careful review. “had roof”… was rewritten as “hard roof”.

  1. Page 3: 3rd line in the second paragraph: “Their study gave meaningful finding” should be deleted. Instead, the finding should just be stated

Reply: Thank you for the good suggestion. We deleted the above inappropriate expression.

  1. Page 3: 5th line in the second paragraph: “Their researches … ” should be rewritten as “Their study …”

Reply: Thank you for the good suggestion. “Their researches … ” was revised as “Their study …”

  1. Page 3: 2nd to the last line in the second paragraph: “Their researches … ” should be rewritten as “Their study …”

Reply: Thank you for the good suggestion. “Their researches … ” was revised as “Their study …”

  1. Page 3: 1st line in the third paragraph: The comma in “The first way is that, the fluid pressure … ” should be deleted. The same comment applies to the 3rd line in the same paragraph that reads “The second way is that, the fluid pressure …”

Reply: Thank you for the good suggestion. We deleted the comma in above expression.

  1. Page 3: Last paragraph: The findings of the study should not be mentioned there. This last paragraph should focus on highlighting the aim and objectives of the study and summarizing how the work addresses the question raised by the research.

Reply: Thank you for the good suggestion. We removed the findings of the study in the last paragraph in the initial paper. In the revised paper, we focus on aim, objectives, question and method in the last paragraph.

  1. Page 4: 5th line under Section 2: “… include the continue …” should be corrected. What is "continue equation"? The same correction should be done on the remaining parts of the paper where this wrong nomenclature is mentioned.

Reply: Thank you for the good suggestion. The continue equation is the mass conservation equation frequently mentioned in fluid mechanism. We added explanation of this expression in the 5th line under Section 2.

  1. Page 5: Last line of the first paragraph: “…. in the discrete form” should be rewritten as “…. in the discrete form given by Eq. (4).”

Reply: Thank you for the good suggestion. we revised the expression according to your suggestion.

  1. Page 6: 2nd line in the paragraph above equation (10): What does the author mean by “continue sine pulse… ”?

Reply: It means that the inlet pressure is applied in the form of sine wave, which is continue in time.

  1. Page 7: The author mentioned in the first paragraph of Page 7 that the work focus on the loading stage of pulse pressure before the breakdown of rock. Comments should be added to justify this choice and also to add some remarks about what the limitation and applicability of this approach are.

Reply: Thank you for the good suggestion. We add some comments to discuss the present choice in the paper.

  1. Page 7: 6th paragraph: the word “we” should be replaced with something more appropriate such as “this study considers …” or something similar.

Reply: Thank you for the good suggestion. We revised the inappropriate expression according to your suggestion.

  1. The study should comment on the implication of such PHF simulation and the pressure behavior on the number of fissures/fractures generated in real life (in the field). Single versus multiple fracture simulation in non-PHF studies showed some remarkable differences in the treatment parameters. The author should add to the manuscript, the following recent paper that discussed this fact on conventional HF and the manuscript should make suggestions as to how these phenomena differ in the PHF.
  • Mukhtar, F. M., Shauer, N., & Duarte, C. A. (2022). Propagation mechanisms and parametric influence in multiple interacting hydraulic fractures: A 3-DG/XFEM hydro-mechanical modeling. International Journal for Numerical and Analytical Methods in Geomechanics.

Reply: Thank you for the good suggestion. The fluid pressure does affect the rupture of the rock, further influencing the number of fissures/fractures. Usually, the number of fissures increases with the pulse peak pressure. Also, the number also affected by other parameters pointed by Mukhtar et al. (2022).

  1. It would benefit the readers if the author can add a new section dedicated to the practical applicability of the simulation so conducted and how its findings can be used as a valuable guide for the PHF design (as previously claimed in the introduction section of the manuscript).

Reply: Thank you for the good suggestion. we added a new discussion in the last paragraph in section 3. The present method shows a huge application potential. In the next step, we plan to design a pulse pump to achieve the pulse supercharging process. There are lots of work to do, such as the design of pulse pump. In fact, the pulse pump is not a traditional water pump. The pulse pump in our study has particular demands of flow rate. We try to design a pump which can be accurately controlled by flow rate.

  1. The English language needs a review by a competent language editor.

Reply: Thank you for the good suggestion. We carefully polished the language, including the words and sentences.

Reviewer 2 Report

Overall, I feel this study presents a relatively decent theoratical work on pulse fracturing. I suggest the authors somehow need to provide how their theoratical results/conclusions are linked to the field practices, and/or what can be implied for future field applications from this work. Also, I think this work can be improved by carefully polishing the techinical writing.  

Author Response

Reply: Thank you for the positive comments and good suggestion. we added a new discussion in the last paragraph in section 3. The present method shows a huge application potential. In the next step, we plan to design a pulse pump to achieve the pulse supercharging process. There are lots of work to do, such as the design of pulse pump. In fact, the pulse pump is not a traditional water pump. The pulse pump in our study has particular demand of flow rate. We try to design a pump which can be accurately controlled by flow rate.